# IWS1 positions downstream DNA to globally stimulate Pol II elongation

Aiturgan Zheenbekova[1,3], James L. Walshe ●[1,3], Moritz Ochmann ●[1], Moritz Bäuerle[1], Ute Neef[1], Kerstin C. Maier[1], Petra Rus[1], Yumeng Yan[2], Henning Urlaub ●[2], Patrick Cramer[1] ✉ & Kristina Žumer ●[1] ✉

The protein IWS1 (Interacts with SPT6 1) is implicated in transcription-associated processes, but a direct role in RNA polymerase (Pol) II function is unknown. Here, we use multi-omics kinetic analysis after rapid depletion of IWS1 in human cells to show that loss of IWS1 results in a global decrease of RNA synthesis and a global reduction in Pol II elongation velocity. We then resolve the cryo-EM structure of the activated Pol II elongation complex with bound IWS1 and elongation factor ELOF1 and show that IWS1 acts as a scaffold and positions downstream DNA within the cleft of Pol II. In vitro assays show that the disordered C-terminal region of IWS1 that contacts the cleft of Pol II is responsible for stimulation of Pol II activity and is aided by ELOF1. Finally, we find that the defect in transcription upon IWS1 depletion leads to a decrease of histone H3 tri-methylation at residue lysine-36 (H3K36me3), but that this secondary effect is an indirect function of IWS1. In summary, our structure-function analysis establishes IWS1 as a Pol II-associated elongation factor that acts globally to stimulate Pol II elongation velocity and ensure proper co-transcriptional histone methylation.

Eukaryotic transcription of messenger RNAs (mRNAs) and many non-coding RNAs by RNA polymerase (Pol) II is a highly regulated process[1–3]. In humans, its misregulation has been linked to various diseases, including developmental disorders, immune response issues, and cancers[4–8]. The longest phase of transcription is elongation, which is a complex process that is also coupled to RNA processing and chromatin modification[9]. Numerous elongation factors that stimulate this complex process have been identified (reviewed in refs. 7,8,10). Some have dual roles, such as DRB sensitivity-inducing factor (DSIF), which together with the negative elongation factor (NELF) initially stabilizes promoter-proximal pausing of Pol II[11,12]. However, after phosphorylation by the positive transcription elongation factor-b (P-TEFb) and release of paused Pol II into productive elongation it is converted to a positive elongation factor that remains associated with elongating Pol II[12–15]. The elongating Pol II is also associated with transcription elongation factor SPT6 and the PAF1 complex (PAF)[13],

which have distinct and multifaceted roles in transcription, including stimulation of elongation velocity, supporting processivity and deposition of histone modifications[16–23]. PAF activity also synergizes with TFIIS, which stimulates the intrinsic endonucleolytic activity of Pol II to overcome backtracking that may occur at transcription barriers[17,24]. Another distinct elongation factor, ELOF1 (Elf1 in yeast)[25] binds Pol II and bridges the Pol II cleft to stimulate transcription[26]. Recent studies have linked ELOF1 to nucleotide excision repair (NER), further highlighting its diverse functional roles[27]. Other factors like the histone chaperone FACT can strongly stimulate transcription indirectly without binding to Pol II[24,28,29]. Taken together, elongation factors stimulate transcription and transcription-associated processes both directly and indirectly through distinct mechanisms.

IWS1 (interacts with SPT6) in humans (Spn1 in yeast) is another putative elongation factor. It was initially identified in three independent studies in yeast that linked it to transcriptional regulation[30–32].

[1]Department of Molecular Biology, Max Planck Institute for Multidisciplinary Sciences, Göttingen, Germany. [2]Bioanalytical Mass Spectrometry, Max Planck Institute for Multidisciplinary Sciences, Göttingen, Germany. [3]These authors contributed equally: Aiturgan Zheenbekova, James L. Walshe. ✉e-mail: patrick.cramer@mpinat.mpg.de; kristina.zumer@mpinat.mpg.de

IWS1 was reported to interact with the phosphorylated C-terminal domain (CTD) of the largest Pol II subunit[33], but recent structural studies have shown that the structured core of yeast Spn1 – comprising a TFIIS N-terminal domain (TND) and HEAT repeat—binds the Pol II elongation complex and is positioned at the tip the Pol II clamp[34] in proximity to yeast Elf1[26]. Similar genomic occupancy and co-purification profiles have been observed for Elf1 and Spn1, suggesting complementary roles[35] for these elongation factors. This is further supported by a single molecule tracking study that suggests Spn1 is a stable component of the elongation complex[36]. In humans and yeast, IWS1 has been implicated in the recruitment of the histone methyltransferase SETD2 to actively transcribed genes and thereby supporting trimethylation of histone H3 at lysine-36 (H3K36me3)[37–40]. IWS1 was shown to directly interact with SPT6 in yeast[41,42] and *Encephalitozoon cuniculi*[43], further underscoring its involvement in transcriptional regulation. Spn1 also interacts with histones and has weak histone chaperone activity in vitro[44–47]. Knockdowns of IWS1 in mouse and human cells have shown that IWS1 plays a role not only in the deposition of H3K36me3, but also in RNA splicing and efficient mRNA export from the nucleus[33,37,38]. In addition, mutation of a conserved unstructured motif in human IWS1 (termed TND interaction motifs—TIMs) has been shown to impair pause release, leading to Pol II accumulation near promoter-proximal regions[48]. In yeast, Spn1 is crucial for normal RNA synthesis[39] and suppressor mutations identified several functionally distinct complexes and activities[49], but whether IWS1 plays a direct role in transcription remains unknown.

Despite these insights, the precise molecular mechanism by which IWS1 promotes transcription elongation remains enigmatic. In our study, we utilized rapid depletion of IWS1 in human cells combined with multi-omics analysis to examine the direct role of IWS1 in Pol II transcription. These in vivo approaches revealed that IWS1 is crucial for normal RNA synthesis by maintaining Pol II elongation velocity. These in vivo analyses were complemented with high-resolution cryo-EM and biochemical assays using highly purified proteins and a chromatinized DNA template to uncover a primary role of IWS1 in supporting transcription elongation and a secondary effect on histone methylation. Collectively, these results establish IWS1 as a key factor in transcription elongation in human cells.

## Results

### IWS1 depletion leads to decreased RNA synthesis

To investigate the primary role of IWS1, we modified the endogenous *IWS1* gene in the human K562 cell line to express IWS1 with an N-terminal dTAG (Fig. 1a, Supplementary Fig. 1a), which allows for rapid depletion of the targeted protein upon binding the dTAG7 ligand[50]. We observe near-complete depletion of IWS1 from cell lysates and chromatin within 1 h of treatment with dTAG7 (Fig. 1b, Supplementary Fig. 1b–c).

To examine the effects of IWS1 depletion on nascent transcription, we performed transient transcriptome sequencing (TT-seq) to assess changes in RNA synthesis following rapid depletion of IWS1 (Supplementary Fig. 1d). TT-seq is based on a short metabolic 4sU-labeling pulse followed by isolation of 4sU-labeled RNA combined with fragmentation to enable measurement of new RNA synthesis[51]. To enable detection of potential global changes in RNA synthesis, we included synthetic RNA spike-ins. We observe that depletion of IWS1 in human cells leads to a significant global reduction in RNA synthesis (Fig. 1c), which is consistent in all gene biotypes. This is further supported by differential gene expression analysis, which reveals that nearly 74% of expressed genes (7504 out of 10147) exhibit a significant reduction in RNA synthesis upon IWS1 depletion (Fig. 1d). This observation is in line with the stimulatory role of the yeast IWS1-homolog, Spn1, on mRNA production[39].

To investigate RNA synthesis in the spatial context of genes, we generated metagene profiles of TT-seq signal aligned at the TSSs and

the transcript end sites (TESs) of genes (Fig. 1e). These metagene profiles show a more distinct decrease in RNA synthesis towards the 3' end regions compared to the 5' ends, which results in a more pronounced downward-tilt in the profile for cells depleted of IWS1 compared to the control cells. To delve deeper into this transcription defect, we categorized expressed genes into four length-based quartiles (shortest, short, long, longest) and generated metagene profiles (Fig. 1f). Although all groups show a similar reduction in RNA synthesis upon depletion of IWS1 (Supplementary Fig. 1e), the metagene profile tilt is less pronounced in short genes and more pronounced in longer genes (compare left panels to right panels, Fig. 1f). An increased downward-tilt in RNA synthesis metagene profiles has been reported previously for elongation defects[20,52]. Taken together, these results show a global stimulatory role of IWS1 in RNA synthesis by regulating transcription elongation and indicate its potential role in maintaining elongation velocity.

### Loss of IWS1 decreases Pol II elongation velocity

To directly investigate the potential role of IWS1 in regulation of elongation velocity, we performed mammalian native elongating transcript sequencing (mNET-seq) to monitor Pol II occupancy (Supplementary Fig. 2a). mNET-seq provides strand-specific, single-nucleotide-resolution data of transcriptionally engaged Pol II[53,54]. Rapid depletion of IWS1 leads to a significant increase in Pol II occupancy in genes (Fig. 2a, Supplementary Fig. 2b), which is consistent for protein-coding and long non-coding RNA genes (Supplementary Fig. 2c). To visualize these changes in a spatial context, we generated a heatmap of changes in mNET-seq signal upon depletion of IWS1 (Fig. 2b, Supplementary Fig. 2d). This shows an accumulation of transcribing Pol II within genes and a decrease downstream of genes. To confirm this, we performed ChIP-seq[55] for Pol II (Supplementary Fig. 2e). We observe a similar increase in signal in genes, which also extends throughout the gene body (Fig. 2c, Supplementary Fig. 2f). The observed accumulation of Pol II in gene bodies is likely due to a slowdown of Pol II within these areas. To further disentangle this, we utilized the ratio between RNA synthesis (TT-seq) and Pol II occupancy (mNET-seq) data as a proxy of Pol II elongation velocity[20,28,56,57] and generated heatmaps to visualize these changes. Our analysis indicates a statistically significant global decrease in the elongation velocity of Pol II upon loss of IWS1 (Fig. 2d–e). Slow elongation rates have been linked to increased splicing efficiency[58–60]. To determine whether we observe changes in splicing upon IWS1 depletion we assessed changes in intron retention rates. In line with the previously reported link between slow elongation and splicing, we found that the loss of IWS1 leads to more efficient splicing (Fig. 2f). Taken together, depletion of IWS1 leads to slower Pol II and its accumulation within gene bodies, similar to depletion of RTF1, an allosteric stimulator of transcription[20,61], suggesting a direct role for IWS1 in transcription.

### IWS1 stimulates Pol II elongation in vitro

To determine whether IWS1 can stimulate Pol II elongation, we performed RNA extension assays in vitro using a minimal, highly purified biochemical system (Supplementary Fig. 3a). Given the overlapping gene occupancy profiles of Spn1 and Elf1[35] and their spatial proximity upon binding Pol II[26], we additionally tested whether ELOF1, the human homolog of Elf1, contributes to IWS1-mediated stimulation. Our experimental setup utilized a chromatinized DNA template containing a single nucleosome and a fluorescently labeled RNA primer (Fig. 3a) that is extended by the activated elongation complex (EC*: Pol II, DSIF, PAF1 complex, SPT6, and TFIIS) and assessed the impact of IWS1 and ELOF1 on Pol II elongation. The transcription reactions were incubated for 30 min and the extended RNA products were then analyzed by denaturing gel electrophoresis (Fig. 3b). We observed two distinct extension products: a full-length transcript (228 nt) resulting from

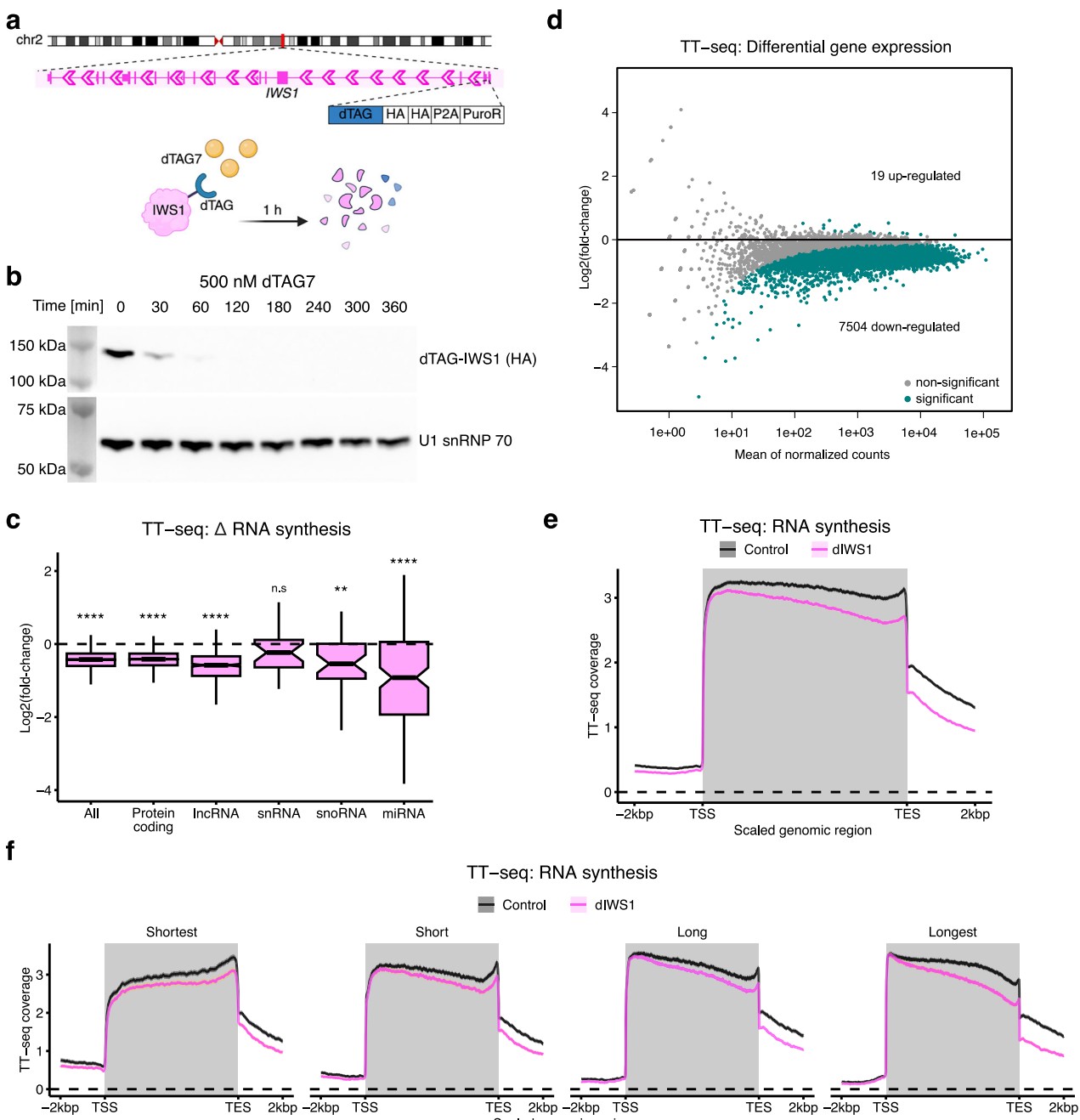

**Fig. 1 | IWS1 depletion leads to decreased RNA synthesis. a** Schematic representation of dTAG insertion into *IWS1* gene using CRISPR/Cas9. Created in BioRender. Zheenbekova, A. (2025) https://BioRender.com/mjvatf1. **b** Western blots of IWS1 (anti-HA) in lysates of dTAG-IWS1 cells treated with dTAG7 for the indicated time. U1 snRNP 70 is a loading control. The experiment was repeated two times from independent biological replicates. **c** Boxplots showing fold-changes (dIWS1/control) in RNA synthesis (TT-seq) in expressed genes (All, $n = 10{,}147$) and individual gene biotypes (protein coding, $n = 9089$; lncRNA, $n = 615$; snRNA, $n = 25$; snoRNA, $n = 41$; miRNA, $n = 145$) after 1 h depletion of IWS1. The thickened line represents the median and the hinges represent the first and third quartiles. The notches stretch to 1.58-times the interquartile range, divided by the square root of the sample size, approximating a 95% confidence interval. Whiskers extend to the maximum and minimum values within 1.5-times the interquartile range from the hinge, outliers are not shown. *p*-values were determined using one-sample two-sided Wilcoxon test (mu = 0), and shown as n.s = $p > 0.05$, *$p < 0.05$, **$p < 0.01$, ***$p < 0.001$, ****$p < 0.0001$. All: $p < 2.2 \times 10^{-16}$, protein-coding: $p < 2.2 \times 10^{-16}$, lncRNA: $p < 2.2 \times 10^{-16}$, snRNA: $p = 0.2411$, snoRNA: $p = 0.007296$, miRNA: $p = 2.116 \times 10^{-9}$. **d** MA plot of TT-seq data after 1 h depletion of IWS1 showing differential gene expression ($n = 10{,}147$). Significantly differentially expressed genes are shown in green and other genes are shown in gray. **e** Metagene analysis of mean TT-seq coverage in expressed genes ($n = 10{,}147$) in 1 h control- or dTAG7-treated (dIWS1) cells. The scaled metagene profiles are aligned at the transcript start (TSS) and transcript end site (TES). The y-axis represents the bootstrapped mean of $\log_2$-transformed normalized TT-seq coverage and the shaded areas around the mean indicate 95% confidence intervals of the mean. The gene region is shaded in gray. **f** As in **e**), but split into gene length quartiles: shortest (<10 kb, $n = 2537$), short (10–26.5 kb, $n = 2537$), long (26.5–65 kb, $n = 2536$), and longest (> 65 kb, $n = 2537$) genes.

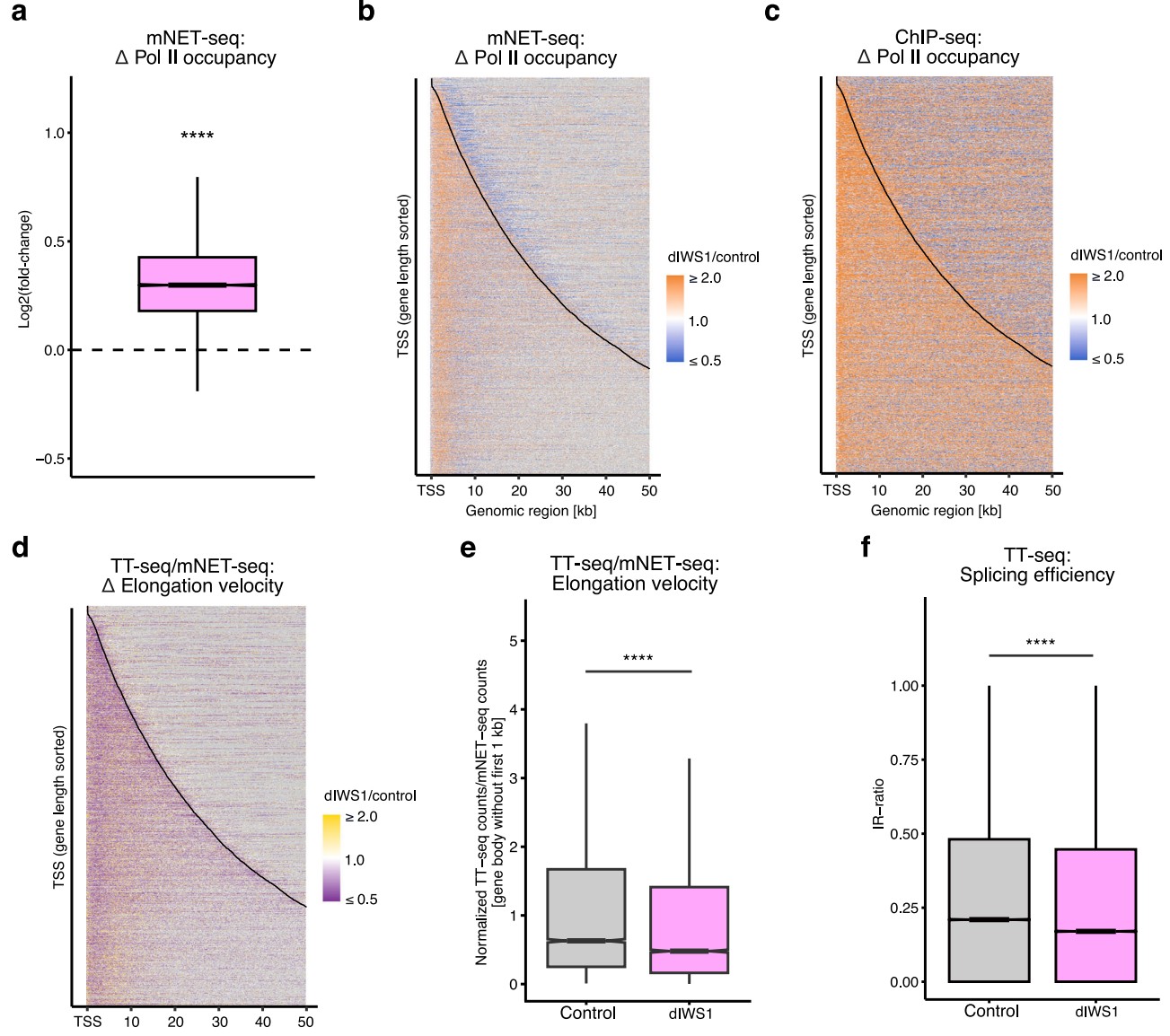

**Fig. 2 | Loss of IWS1 decreases Pol II elongation velocity. a** Boxplot showing fold-changes (dIWS1/control) in Pol II occupancy (mNET-seq) after 1 h depletion of IWS1. Representations as in Fig. 1c. $p < 2.2 \times 10^{-16}$. **b** Heatmap representation of changes in mNET-seq signal in expressed genes ($n = 10{,}147$) after depletion of IWS1. Gene regions are aligned at TSS. The black line represents ends of the genes and bins without signal are shown in gray. **c** As in (**b**), but for Pol II ChIP-seq signal.

**d** Heatmap representation of changes in Pol II elongation velocity after 1 h depletion of IWS1. Representation as in (**b**). **e** Boxplots show mean TT-seq/mNET-seq ratios per gene ($n = 10{,}147$). $p$-values were determined using two-sample two-sided Wilcoxon test, and indicated as follows: n.s = $p > 0.05$, *$p < 0.05$, **$p < 0.01$, ***$p < 0.001$, ****$p < 0.0001$. Representation as in Fig. 1c. $p < 2.2 \times 10^{-16}$. **f** Boxplots of intron retention ratios (IR-ratio) per gene in TT-seq. Representation as in **e**. $p < 2.2 \times 10^{-16}$.

completed nucleosome passage, and an intermediate transcript (~83 nt), corresponding to the position where Pol II first encounters the nucleosome[28]. The addition of IWS1 increases nucleosome passage and extension to full-length transcripts, whereas inclusion of ELOF1 does not have a significant effect, contrary to previous observations in yeast[26]. Notably, the inclusion of both ELOF1 and IWS1 further increases nucleosome passage and suggests a functional cooperation between these two factors (Fig. 3b–c). This finding, alongside prior observations that RTF1 also stimulates transcription through nucleosomes[24], supports a model in which IWS1 directly enhances Pol II elongation.

### Architecture of the activated elongation complex with IWS1 and ELOF1

To elucidate the role of ELOF1 and IWS1 in the stimulation of Pol II elongation through nucleosomes, we reconstituted the complete

mammalian activated elongation complex (based on ref. 13) on a nucleosome template in the presence of TFIIS, IWS1, and ELOF1. We used a modified Widom-601 nucleosome positioning sequence to stall the elongation complex 32 bp into the nucleosome and purified this complex by size exclusion chromatography (Supplementary Fig. 3b–c). Purified complexes were crosslinked and flash-frozen for single-particle cryo-EM analysis (Supplementary Fig. 4a–b). Cryo-EM reconstruction yielded a 2.5 Å resolution map of Pol II, and focused 3D classification with signal subtraction located all elongation factors except TFIIS and RTF1, enabling high-fidelity model building (Fig. 4b–f, Supplementary Figs. 4–6). A partially unwrapped downstream nucleosome was observed in a small subset of particles and was therefore not modeled (Supplementary Fig. 4). Pol II adopts a post-translocated state, at the designed stall position (Supplementary Fig. 10b). Consistent with previous observations, the structured core of IWS1 is located between the clamp head of Pol II and the KOW2 domain

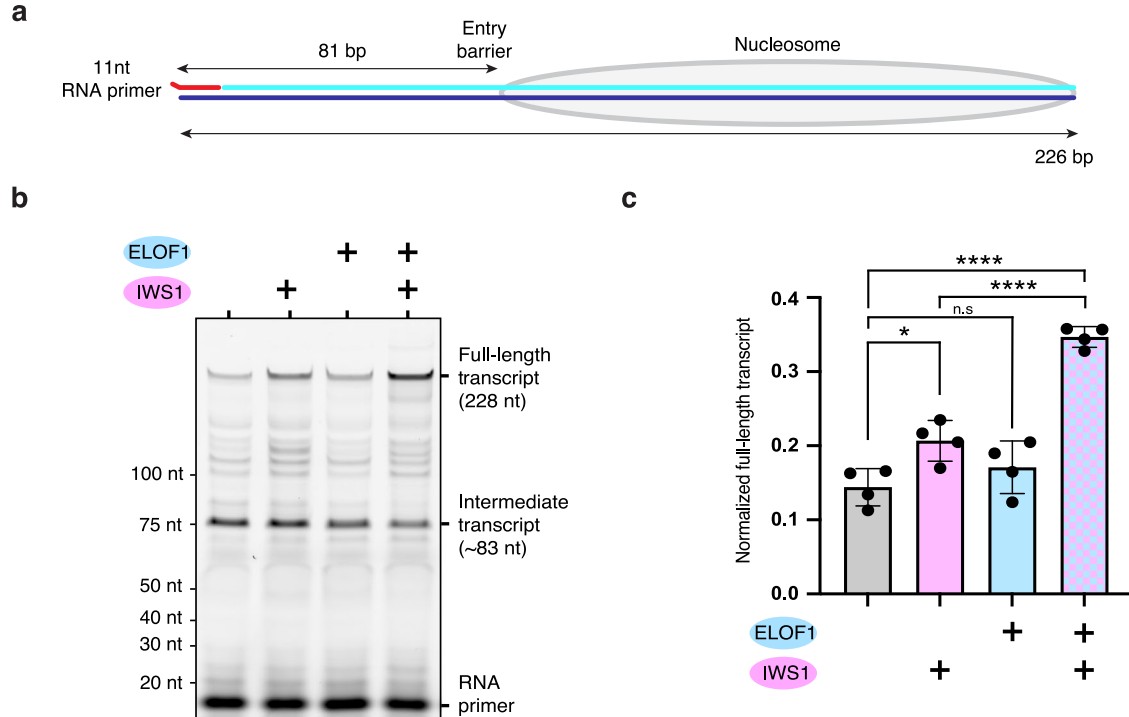

**Fig. 3 | IWS1 stimulates Pol II elongation in vitro. a** Schematic representation of the nucleosome-positioning DNA template used for RNA extension assays. RNA primer is shown in red, the template and non-template DNA strands are shown in blue and cyan, respectively. **b** Denaturing PAGE analysis of RNA extension assays. Fluorescently labeled RNA primer was extended by the activated elongation complex ± IWS1 ± ELOF1. RNA lengths and the main extension products are indicated on the left and right side, respectively. **c** Quantification of the full-length transcripts from the RNA extension assay shown in **b**). Data represent mean ± SEM from four independent replicates. *p*-values were determined using one-way ANOVA followed by Tukey's multiple comparisons test, and indicated as follows: n.s = $p > 0.05$, *$p < 0.05$, **$p < 0.01$, ***$p < 0.001$, ****$p < 0.0001$. Statistical significance was observed between three conditions (+/− IWS1 ($p = 2.7 \times 10^{-2}$), +/− both IWS1 and ELOF1 ($p = 8.4 \times 10^{-7}$) and +/− ELOF1 with IWS1 present ($p = 4.1 \times 10^{-5}$). No significance was observed between +/− ELOF1 alone ($p = 0.5$).

of SPT5, whilst ELOF1 binds the lobe of RPB2 and bridges the cleft of Pol II (Fig. 4a–b)[10,26,27,34].

**Elongation Complex Scaffold domain.** In our data, we observe four previously unreported interaction sites between IWS1 and the elongation complex. Collectively we term these regions of IWS1 the Elongation Complex Scaffold (ECS) domain (Fig. 4a). Interaction Site 1: A small helix of IWS1 (residues 528-539), N-terminal to the IWS1 core, contacts the jaw of RPB5 below the DNA, bridging the lower jaw and clamp of Pol II (Supplementary Fig. 7a). Interaction Site 2: C-terminal to the IWS1 core, residues 649–711 form a tri-partite interface with the TIM of SPT6 and the NGN domain of SPT5 (Fig. 4d), suggesting a role for IWS1 in stabilizing elongation factor binding to Pol II. Interaction Site 3: IWS1 residues 748–770 traverse above the RPB2 lobe, extend past SPT5 to wrap around the zinc finger of ELOF1 (Fig. 4e, Supplementary Fig. 7d–e). This interaction positions two conserved arginine residues of IWS1 (R751 and R753) to engage with the non-template (NT) DNA strand, alongside W250 of SPT5, that participates through π–π stacking, to collectively reinforce the open transcription bubble (Fig. 4e, Supplementary Fig. 7c). This configuration would sterically occlude Cockayne syndrome A (CSA) ubiquitin ligase from binding to ELOF1 (Supplementary Fig. 8a). The extensive IWS1–ELOF1 interactions explain the observed synergy between ELOF1 and IWS1 in our RNA extension assays (Fig. 3b–c). Interaction Site 4: The extreme C-terminal region of IWS1 (residues 786–819) inserts into the Pol II cleft via the RPB1 jaw domain (Fig. 4c, f). This interaction bridges the assembly and jaw sub-domains of RPB5 and terminates at the clamp of RPB1, effectively wrapping the downstream DNA. IWS1 binding is stabilized by conserved hydrophobic residues and basic residues, that orientate toward the DNA (Supplementary Fig. 9). Notably, in particles lacking

ELOF1 but retaining the IWS1 core, this C-terminal insertion is absent (Supplementary Fig. 7b; compare MAP2–3), confirming the requirement of ELOF1 for IWS1 C-terminal engagement with Pol II. Critically, the presence of the IWS1 C-terminus within the Pol II cleft restricts the movement of the downstream DNA (Supplementary Fig. 5, 10a, c–d). Collectively, the four interaction sites of the ECS domain facilitate elongation factor binding, reinforce the open transcription bubble, and position the downstream DNA within the Pol II cleft.

## C-terminus of IWS1 ECS domain stimulates Pol II elongation in vitro

To determine if the ECS domain of IWS1 is responsible for IWS1-mediated stimulation of Pol II elongation, we tested a series of IWS1 truncations in the presence of ELOF1 (Fig. 5a). All mutants contain the structured IWS1 core, which by itself does not stimulate Pol II elongation (T1) (Fig. 5b–c). Inclusion of the N-terminal region of IWS1 is not sufficient to rescue stimulation (T2), indicating the essential role of the ECS in promoting Pol II elongation. Truncation 3 (T3) that includes Interaction Sites 1-3, and only lacks the final 40 residues of the IWS1 ECS domain, also does not stimulate transcription. Only Truncation 4 (T4) that includes the C-terminal region of the ECS domain (Interaction Sites 2-4), but lacks the N-terminal region, stimulates Pol II elongation and nucleosome passage, albeit to a lesser extent than the wild-type protein. Taken together, these data confirm our structural observations that the C-terminal residues of Interaction Site 4 in the IWS1 ECS domain can directly stimulate Pol II elongation. In addition, the N-terminal region of IWS1, previously shown to bind histones[45] and to contain three TIMs[48], does not stimulate elongation without the ECS, further supporting a direct, rather than chaperone-mediated indirect mechanism of action of IWS1.

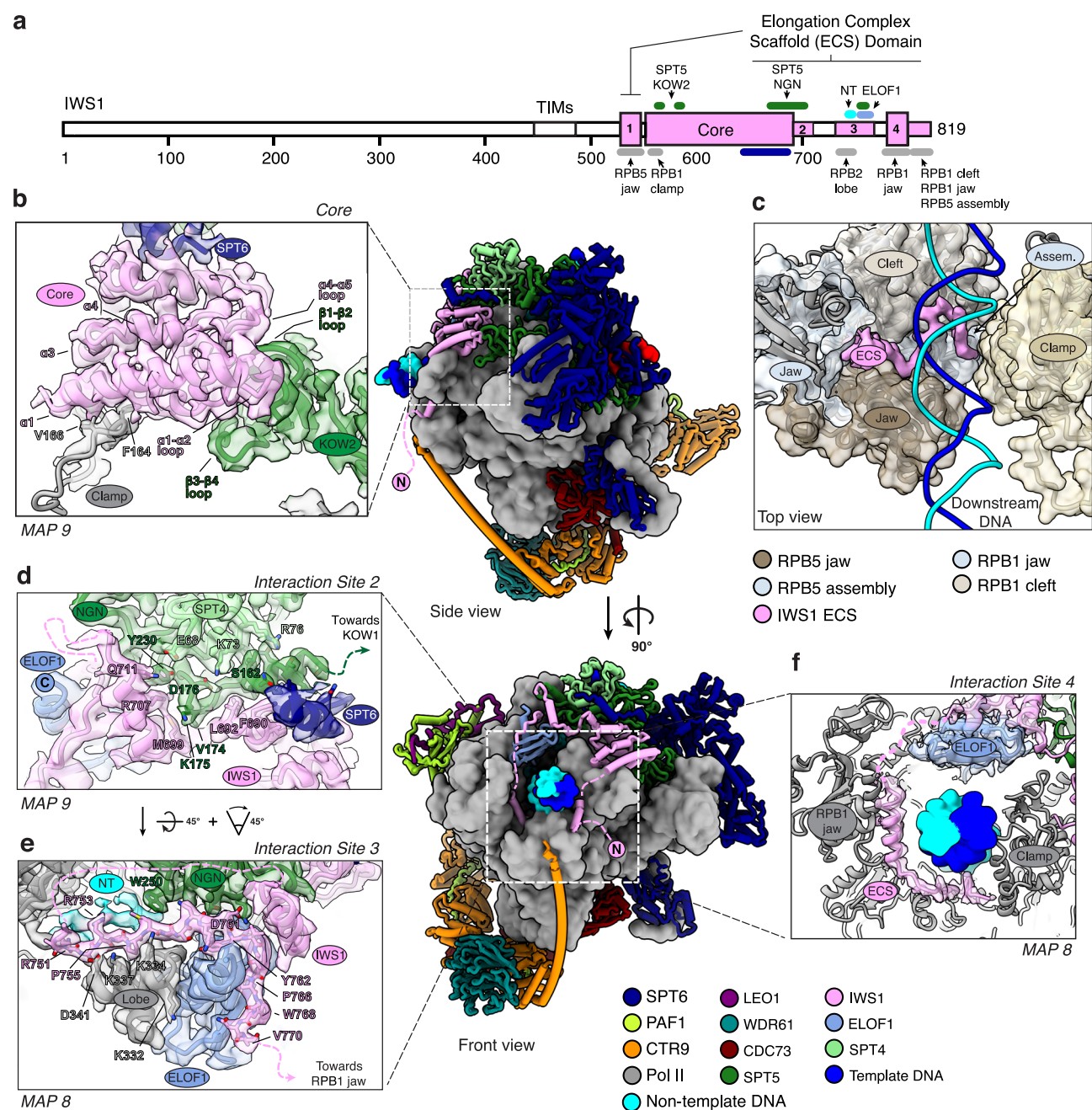

**Fig. 4 | Architecture of the activated elongation complex with IWS1 and ELOF1. a** Schematic 2D overview of IWS1 (top). Colored regions represent residues of IWS1 that we resolved in cryo-EM reconstructions. IWS1 binding regions for Pol II, elongation factors (SPT5, SPT6 and ELOF1), and the non-template DNA (NT) strand are indicated in their respective colors. IWS1 regions corresponding to Interaction Sites 1-4 are numbered and the regions making up the Elongation Complex Scaffold (ECS) domain are indicated. The side and front views of the overall structure of the activated elongation complex with IWS1 and ELOF1 (bottom). Pol II is depicted in gray and the DNA and elongation factors are depicted in their respective colors. **b** Zoomed-in panel highlighting the interaction between IWS1 core and RPB1 clamp head and KOW2 domain of SPT5, respectively. **c** Top view of the Pol II cleft showing the C-terminal region of the ECS domain of IWS1. The ECS C-terminus binds to the RPB1 jaw, bridges between the RPB5 jaw and assembly domains, and extends to the RPB1 clamp. Pol II domains are shown in cartoon and transparent surface representations, colored according to the legend. The cryo-EM density for the ECS domain of IWS1 from *MAP8* is shown. **d** Interaction Site 2: Residues 649–711 of the ECS domain of IWS1 makes a three-way contact between the NGN domain of SPT5 and the TND interaction motif (TIM) of SPT6. **e** Interaction Site 3: Residues 748–770 of the ECS domain of IWS1 interact with the non-template strand of the open transcription bubble (NT), bind the RPB2 lobe, and interact with zinc finger of ELOF1. **f** Interaction Site 4: Residues 786–819 of the ECS domain of IWS1 bind to the RPB1 jaw and cleft domains. Cryo-EM densities in (**b**, **d**–**h**) are rendered as transparent surfaces and both densities and cartoons are colored as per legend.

## Defects in transcription elongation lead to decreased H3K36me3

The combined multi-omics, cryo-EM and biochemical data outlined above reveal a primary role for IWS1 in the stimulation of Pol II elongation. IWS1 has also been implicated in the recruitment of the histone methyltransferase SETD2 to actively transcribed genes to support co-transcriptional H3K36me3 deposition[37,40]. However, recent crosslinking-mass spectrometry data of highly purified in vitro reconstituted complexes detected limited stable IWS1-SETD2 contacts[62], suggesting the recruitment of SETD2 by IWS1 may be a

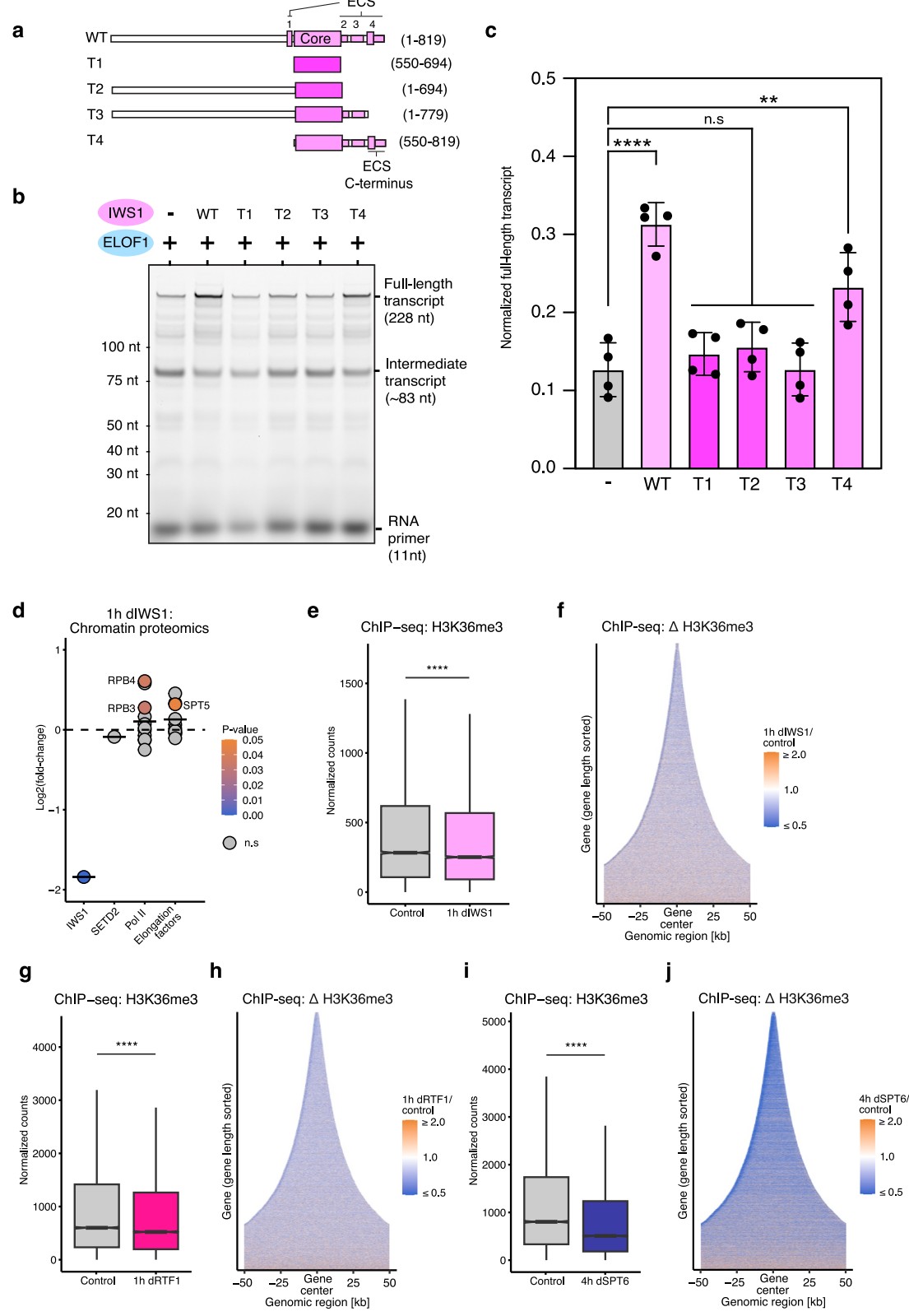

secondary effect. To test whether IWS1 recruits SETD2, we performed proteomic analysis of chromatin-associated proteins 1 and 4 h after IWS1 depletion (Fig. 5d, Supplementary Fig. 11a, Supplementary Data 1). Contrary to previous studies, the level of chromatin-associated SETD2 did not change upon IWS1 depletion (Fig. 5d, Supplementary Fig. 11a)[37,39,40]. On the other hand, Pol II and transcription elongation factors increased, consistent with the increase in Pol II occupancy we

observe (Fig. 2a–c). Surprisingly, ChIP-seq analysis shows a reduction in H3K36me3 levels in genes following 1h IWS1 depletion (Fig. 5e, Supplementary Fig. 11b–c) that intensifies when the depletion is extended to 4 h (Supplementary Fig. 11d–g). Heatmaps of changes in H3K36me3 ChIP-seq signal aligned at gene centers reveal a clear decrease in H3K36me3 levels across gene bodies after IWS1 depletion (Fig. 5f, Supplementary Fig. 11f). Taken together these results suggest

**Fig. 5 | ESC domain of IWS1 stimulates Pol II elongation to support H3K36me3.**
**a** Schematic 2D overview of IWS1 truncation mutants analyzed in RNA extension assays (T1-4). **b** Denaturing PAGE analysis of RNA extension assays. Fluorescently labeled RNA primer was extended by the activated elongation complex with ELOF1 ± IWS1 mutants. Labels as in Fig. 3b. **c** Quantification of the full-length transcripts from the RNA extension assay shown in **a**). Data represent mean ± SEM from four independent replicates. *p*-values were determined using one-way ANOVA followed by Tukey's multiple comparisons test, representation as in Fig. 3c. Statistical significance was observed between multiple conditions including +/− IWS1 WT ($p = 6.2 \times 10^{-7}$) and +/− IWS1 T4 ($p = 3.5 \times 10^{-3}$). No significance was observed between +/− T1, T2 or T3 ($p = 0.9$, $p = 0.9$, $p = 0.9$, respectively). **d** Dotplot depicts fold changes in levels of Pol II and its associated factors in the chromatin fraction

after 1 h depletion of IWS1. Each dot represents a factor/subunit, where colors denote *p*-value, non-significant changes are shown in gray. The horizontal line represents mean log2 fold-change of each set. *p*-values were assessed using two-sided Student's *t*-test. **e** Boxplots show H3K36me3 ChIP-seq counts of expressed genes ($n = 10,147$) in 1 h control- or dTAG7-treated (dIWS1) cells. Representation as in Fig. 2e. *p*-value = $1.518 \times 10^{-12}$. **f** Heatmap representation of changes in H3K36me3 ChIP-seq signal of expressed genes ($n = 10,147$) aligned at the gene centers after 1 h depletion of IWS1. Representation as in Fig. 2b. **g** As in **e**), but after 1 h depletion of RTF1 ($n = 12,976$). $p < 2.2 \times 10^{-16}$. **h** As in (**f**), but after 1 h depletion of RTF1 ($n = 12,976$). **i** As in (**e**), but after 4 h depletion of SPT6 ($n = 14,002$). $p < 2.2 \times 10^{-16}$. **j** As in (**f**), but after 4 h depletion of SPT6 ($n = 14,002$).

that the decrease in H3K36me3 following IWS1 depletion is not due to impaired recruitment of its histone methyltransferase SETD2.

Instead of recruiting SETD2, IWS1 may act to stimulate deposition of H3K36me3 by SETD2. To investigate this possibility, we performed in vitro methylation assays, in the context of a transcribing polymerase[62] and examined the effect of including IWS1 on H3K36me3 by SETD2 (residues 1446–2564) (Supplementary Fig. 11h). In the absence Pol II elongation, we observe low levels of H3K36me3, which strongly increase when NTPs are included in the reaction (Supplementary Fig. 11i). In contrast, inclusion of IWS1 leads to a small increase in H3K36me3 in the absence of transcription and also when NTPs are included. These results, in combination with previous work[62], demonstrate that IWS1 neither recruits SETD2 to chromatin nor stimulates its H3K36me3 activity. Our biochemical assay reveals transcription has a greater effect on H3K36me3 than the presence of IWS1. In support of this, we observe that the most affected genes after IWS1 depletion, based on our TT-seq data, show a stronger reduction in H3K36me3 than the least affected genes (Supplementary Fig. 11j). This suggests the in vivo changes in H3K36me3 levels upon IWS1 depletion are a secondary effect of altered transcriptional activity.

To further explore the link between transcriptional elongation defects and H3K36me3, we examined H3K36me3 levels following rapid depletion of two key elongation factors, RTF1 and SPT6[20]. ChIP-seq analysis shows that depletion of both factors reduces H3K36me3 levels within gene bodies (Fig. 5g–j, Supplementary Fig. 11k–m). Notably, SPT6 depletion leads to a stronger decrease in H3K36me3 compared to IWS1 depletion, consistent with its greater impact on Pol II activity[20] and role in positioning SETD2 in proximity to the transcribed nucleosome during elongation[62–64]. The change in H3K36me3 levels upon rapid depletion of RTF1 is more moderate and comparable to IWS1 depletion (Supplementary Fig. 11n), aligning with the similarity in the observed transcriptional defects upon loss of these factors. Taken together, our results indicate that IWS1 has a secondary role in H3K36me3, by sustaining the transcription elongation kinetics and thereby supporting optimal histone methylation.

## Discussion

In this study, we define a primary role of IWS1 as a direct stimulator of transcription elongation by mammalian Pol II. Using in vivo multiomics analysis, biochemical assays, and high-resolution cryo-EM analysis, we show that IWS1 interacts directly with Pol II to enhance RNA synthesis and Pol II elongation velocity. This work separates the primary function of IWS1 in transcription elongation from secondary effects on co-transcriptional H3K36me3 deposition, providing mechanistic insights beyond previous observations[33,37,39,40,48].

Our cryo-EM and biochemical analysis demonstrates that the C-terminal of the ECS domain of IWS1 is responsible for its stimulation of transcription and acts to position the downstream DNA within the cleft, limiting alternate conformations (Supplementary Fig. 10a, c). This architecture may promote forward translocation and is consistent with a model where IWS1 stabilizes an unwound state of the downstream DNA (Supplementary Fig. 10d). Notably, IWS1 binding to the Pol

II jaw overlaps with binding sites for RECQL5 and UVSSA (Supplementary Fig. 8b), suggesting mutually exclusive interactions consistent with the opposing functions of these factors[27,65,66].

In yeast, Elf1 facilitates nucleosome passage via its basic N-terminal region, thought to promote dissociation of histone-DNA contacts[26]; however, in our structure, the mammalian ELOF1 N-terminus faces inward towards the transcription bubble (Supplementary Fig. 7e) and we do not observe a similar stimulation of nucleosome passage by ELOF1 alone, suggesting its role may be tied to IWS1 in higher eukaryotes. The N-terminal region of IWS1 contains multiple TIMs that bind elongation factors, and their mutation impairs pause release in a subset of genes, leading to promoter-proximal Pol II accumulation[48]. However, our biochemical assays demonstrate that the unstructured C-terminus of IWS1 predominantly supports Pol II elongation in vitro, consistent with our structural findings. These observations suggest a multifaceted role of IWS1 in regulation of Pol II elongation.

Finally, contrary to earlier models proposing that IWS1 recruits SETD2 to promote H3K36me3 deposition[37,40], we find that IWS1 depletion does not affect SETD2 recruitment. Instead, the reduction in H3K36me3 is secondary to impaired transcription. Prior studies[37,40] using long depletions with siRNA and lysate-based protein interaction assays likely conflated primary and secondary effects. Our findings align with emerging models in which transcription-coupled nucleosome reassembly governs SETD2-mediated H3K36me3 deposition[62–64]. Consistent with this, depletion of other elongation factors, such as RTF1 and SPT6, similarly reduce H3K36me3 reinforcing the close link between transcription elongation and chromatin modification. Together, our results establish that the primary role of IWS1 is to facilitate transcription elongation by acting in coordination with ELOF1 to position the DNA within the elongating Pol II complex, while the effects on H3K36me3 are secondary to impaired transcription.

## Methods

### Cell line and cell culture

K562 human cells were sourced from the DSMZ-German Collection of Microorganisms and Cell Cultures GmbH. K562 cells and genome edited cell lines were cultured in RPMI medium (Thermo Fisher Scientific), enriched with 10% fetal bovine serum and 1x GlutaMax, within a humidified incubator at 37 °C and 5% $CO_2$. These cells underwent subculturing every 2–3 days, maintaining a density of 150,000 to 700,000 cells/mL. To deplete dTAG-IWS1, the culture medium was supplemented with 500 nM of dTAG7 ligand, previously dissolved in dimethyl sulfoxide (DMSO) at 10 mM concentration. Control cells were treated with solvent only with the same treatment time.

Yeast *S. cerevisiae* cells (BY4741 strain) were procured from Euroscarf (ACC-Y00000). Cells were cultured in yeast medium (YP supplemented with 2% glucose), at 30 °C until the optical density at 595 nm (OD595) reached 0.8.

For H3K36me3 ChIP-seq experiments after SPT6 and RTF1 depletions, SPT6-dTAG and RTF1-dTAG K562 human cell lines were obtained from ref. 20.

## Genome editing with CRIPSR/Cas9

The degradation tag (dTAG) was cloned and integrated into the target protein as previously described[20]. The specific oligonucleotide sequences utilized in this process are detailed in Supplementary Table 1. To confirm the successful integration of the dTAG into the *IWS1* locus, clones were validated by western blotting, using both a HA epitope-specific antibody to detect the dTAG and a primary anti-IWS1 antibody, and Sanger sequencing of the PCR-amplified integration site. These techniques ensured the precise characterization of the dTAG insertion within the *IWS1* gene.

## Western blots for cell lysates

Following treatment, the cells were lysed using RIPA buffer (25 mM Tris-HCl, pH 8.0, 150 mM NaCl, 1% NP-40, 1% sodium deoxycholate, 0.1% SDS) supplemented with 1x protease inhibitors (Sigma-Aldrich), 2 mM $MgCl_2$, and 500 U/mL Benzonase (Sigma-Aldrich). The lysates were incubated on ice for 20 min to ensure thorough cell lysis. Following incubation, the lysates were centrifuged at 15,000 rpm for 15 min at 4 °C. The clear supernatant was then transferred to new tubes for further analysis.

Protein concentrations in the lysates were quantified using the Bradford assay (Bio-Rad), according to the manufacturer's guidelines. Subsequently, 20 μg of protein per sample was loaded into wells and separated with denaturing gradient gel electrophoresis (4–20% Bis-Tris Nupage, Life Technologies) and transferred onto nitrocellulose membranes (GE Healthcare Life Sciences) for Western blotting, which was carried out following standard protocols.

For subcellular fractionation analyses, we employed a fractionation protocol[67] to fractionate 10 million cells. Equal amounts of each fraction were analyzed as described above for whole cell lysates. The antibodies used for western blotting are anti-HA (Roche, REF: 11867431001), anti-IWS1 (Proteintech, REF: 16943-1-AP), anti-H3 (Abcam, REF: ab21054) and anti-U1-snRNP70 (SCBT, REF: sc-390899). Uncropped blots of Fig. 1b, Supplementary Fig. 1b–c are provided in Source Data Fig. 1b, Source Data Supplementary Fig. 1b–c, respectively.

## Protein expression and purification

Pol II and elongation factors used in in vitro assays were purified as previously described[13,24,28,61], IWS1 was expressed in *Trichoplusia ni* (Hi5 insect cells) as an N-terminal, 6xHis MBP fusion protein. Cells were collected and resuspended in lysis buffer (300 mM NaCl, 20 mM Na-HEPES, pH 7.4, 30 mM imidazole, 10% (v/v) glycerol, 1 mM DTT, 0.284 μg ml⁻¹ leupeptin, 1.37 μg ml⁻¹ pepstatin A, 0.17 mg ml⁻¹ PMSF and 0.33 mg ml⁻¹ benzamidine), and flash-frozen in liquid nitrogen and stored at −70 °C. IWS1 was purified from 1 L of Hi5 lysate at 4 °C. The cells were thawed, lysed by sonication and cleared by centrifugation and filtration through 0.8 μm syringe filters. The clarified cell lysate was applied to two 5 mL HisTrap HP columns (Cytivia) equilibrated with lysis buffer. The columns were washed with 3 column volumes of lysis buffer, high salt buffer (1 M NaCl, 20 mM Na-HEPES, pH 7.4, 30 mM imidazole, 10% (v/v) glycerol, 1 mM DTT, 0.284 μg ml⁻¹ leupeptin, 1.37 μg ml⁻¹ pepstatin A, 0.17 mg ml⁻¹ PMSF and 0.33 mg ml⁻¹ benzamidine) and lysis buffer again. IWS1 was eluted from the column with a 10 column volume gradient of nickel elution buffer (300 mM NaCl, 20 mM Na-HEPES, pH 7.4, 500 mM imidazole, 10% (v/v) glycerol, 1 mM DTT, 0.284 μg ml⁻¹ leupeptin, 1.37 μg ml⁻¹ pepstatin A, 0.17 mg ml⁻¹ PMSF and 0.33 mg ml⁻¹ benzamidine). Eluted fractions were analyzed by denaturing polyacrylamide gel electrophoresis (SDS-PAGE). Fractions containing IWS1 were pooled, $MnCl_2$ added (1 mM) and mixed with TEV protease and lambda phosphatase and dialyzed against dialysis buffer (300 mM NaCl, 20 mM Na-HEPES, pH 7.4, 30 mM imidazole, 1 mM $MnCl_2$, 10% (v/v) glycerol, 1 mM DTT, 0.284 μg ml⁻¹ leupeptin, 1.37 μg ml⁻¹ pepstatin A, 0.17 mg ml⁻¹ PMSF and 0.33 mg ml⁻¹ benzamidine). The dialyzed sample was applied to two 5 mL HisTrap HP columns followed by one HiTrap Q column (Cytivia). The columns

were washed with 1 column volume of lysis buffer and the cleaved IWS1 recovered from the flow-through, concentrated in a 50 kDa MWCO Amicon Ultra Centrifugal Filter (Merck) and applied to a HiLoad S200 16/600 pg column equilibrated with gel filtration buffer (300 mM NaCl, 20 mM Na-HEPES, pH 7.4, 10% (v/v) glycerol, 1 mM DTT). Peak fractions were assessed by SDS-PAGE analysis. IWS1 fractions were pooled and concentrated in a 50 kDa MWCO Amicon Ultra Centrifugal Filter (Merck), aliquoted, flash frozen in liquid nitrogen and stored at −70 °C.

SETD2 (residues 1446–2564) were expressed in Hi5 insect cells as a 6xHis MBP fusion protein and sonication and clarification were performed as described for IWS1. Clarified lysate was applied to a XK 16/20 column packed with amylose resin (Cytivia). The column was washed with 3 column volumes of lysis buffer (300 mM NaCl, 20 mM Na-HEPES, pH 7.4, 30 mM imidazole, 10% (v/v) glycerol, 1 mM DTT, 0.284 μg ml⁻¹ leupeptin, 1.37 μg ml⁻¹ pepstatin A, 0.17 mg ml⁻¹ PMSF and 0.33 mg ml⁻¹ benzamidine) and SETD2 eluted with 2 column volumes of elution buffer (lysis buffer supplemented with 10 mM maltose). Eluted fractions were analyzed by SDS-PAGE and fractions containing SETD2 were pooled and incubated with TEV protease for a minimum of 12 h. The cleaved sample was applied to two 5 mL HisTrap HP to remove cleaved MBP-tag and TEV protease. The columns were washed with 1 column volume of lysis buffer and the cleaved SETD2 recovered from the flow-through, concentrated in a 50 kDa MWCO Amicon Ultra Centrifugal Filter (Merck) and applied to HiLoad S200 16/600 pg column equilibrated with gel filtration buffer (300 mM NaCl, 20 mM Na-HEPES, pH 7.4, 10% (v/v) glycerol, 1 mM DTT). Peak fractions were assessed by SDS-PAGE analysis. SETD2 fractions were pooled and concentrated in a 50 kDa MWCO Amicon Ultra Centrifugal Filter (Merck), aliquoted, flash frozen in liquid nitrogen and stored at −70 °C.

ELOF1 was expressed and purified as described previously[27].

## Preparation of mono-nucleosomal constructs

DNA constructs for mono-nucleosomal templates for cryo-EM analysis and RNA extension assays were generated by 50 mL PCR. The template for amplification were vectors containing modified Widom 601 nucleosome positioning sequences with a T-less cassette on the template strand (Supplementary Table 1). PCR reactions were performed with two primers as outline in Supplementary Table 1. The PCR products were purified by anion chromatography exchange, and digested by TspRI. The concentrations of nucleosome DNA constructs were calculated by the extinction coefficient and the absorption at 280 nm. Nucleosome core particle reconstitution was then performed using the salt-gradient dialysis method[68]. The resulting nucleosome was concentrated to 5–10 μM and stored up to 2 weeks at 4 °C.

## Preparation of tetra-nucleosomes

Tetra-nucleosomes were prepared as described previously[62].

## RNA extension assay

In vitro transcription assays with the elongation complex (EC*: Pol II, DSIF, PAF1 complex, SPT6), were performed in the presence of TFIIS and ± IWS1 and ± ELOF1 on a DNA template containing a single nucleosome similar to previously described[24,69]. The concentrations listed refer to the final concentrations used in the assay and reactions were performed in quadruplex. RNA primer (240 nM) was incubated with the nucleosome template (120 nM) for 10 min on ice. DNA template and RNA primer sequences are detailed in Supplementary Table 1. *S. scrofa* Pol II (225 nM) was added and the reactions were incubated for a further 10 min on ice. DSIF (337 nM), SPT6 (337 nM), PAF1 complex (337 nM), P-TEFb (337 nM), and ATP (1 mM) were added along with IWS1 (600 nM), ELOF1 (600 nM) or buffer as indicated. Reactions were incubated for 15 min at 30 °C and transcription started by the addition of CTP, GTP, UTP (0.4 mM each) and TFIIS (45 nM). The

final reaction conditions in 15 µL were 100 mM NaCl, 20 mM Na-HEPES, pH 7.4 at 20 °C, 4 mM $MgCl_2$, 1 mM DTT, and 4.15% glycerol. 6 µL samples were mixed with 0.75 µL 10x $CaCl_2$ buffer (100 mM Na-HEPES pH 7.4 (@ 20 °C), 50 mM $CaCl_2$, 25 mM $MgCl_2$) and proteinase K (20 µg). The samples were incubated for 20 min at 55 °C, cooled down to 37 °C, before 1.5 U DNase I (RNase-free, Thermo Scientific™) were added. The samples were incubated at 37 °C for a further 20 min before the addition of 8 µL 2x unfolding buffer (6.4 M urea, 50 mM EDTA, pH 8.0, 2x TBE buffer) and denatured for 10 min at 95 °C and separated by 12% denaturing PAGE gels (12% acrylamide:bis-acrylamide 19:1, 8 M urea, 1x TBE buffer, ran at 300 V in 0.5x TBE buffer for approx. 30 min)[28].

In vitro transcription assays with IWS1 truncations were performed in an identical manner in the presence of ELOF1.

## Quantification of full-length product by RNA extension assay

Full-length RNA transcripts were quantified using Fiji's integrated density values[70]. Rectangular boxes were placed centered on the full-length transcript and the raw intensity measured using the build-in function. To control for non-transcription related variation, full-length transcript intensities were normalized against the sum of all transcribed product in the corresponding lane, excluding excess RNA primer. All box sizes used in Fiji and the respective raw values can be found in Supplementary Data 2. Graphs were rendered in Prism 9 (GraphPad Software).

## Sample preparation for cryo-EM analysis

The complete activated elongation complex (based on ref. 61) in the presence of TFIIS, IWS1 and ELOF1 was assembled on a nucleosome template in a final buffer containing 130 mM NaCl, 20 mM Na-HEPES pH 7.4 (@ 20 °C), 6 mM $MgCl_2$, 1 mM DTT, and 5% (v/v) glycerol. The reaction volume was 100 µL. A 5′ Cys-5 labeled RNA primer (1.35 µM) (Supplementary Table 1) and the nucleosomal template (0.75 µM) (Supplementary Table 1) were mixed with on ice for 10 min. *S. scrofa* Pol II (0.9 µM) was added and the reaction was incubated for a further 10 min on ice. DSIF (1.8 µM), SPT6 (1.8 µM), IWS1 (4.5 µM), RTF1 (1.8 µM), PAF1 complex (1.8 µM), P-TEFb (1 µM), ELOF1 (4.5 µM), 3′ dATP (1 mM), compensation buffer, water was added and the sample incubated at 30 °C for 15 min. TFIIS (1.8 µM) and GTP, CTP, UTP (0.5 mM each) were added and the extension of the RNA primer allowed to proceed for 30 min at 30 °C. After centrifugation, the sample was purified by gel filtration using a Superose 6 Increase (3.2/300) pre-equilibrated with 75 mM NaCl, 20 mM Na-HEPES pH 7.4 (@ 20 °C), 3 mM $MgCl_2$, 1 mM DTT, and 4% glycerol (Supplementary Fig. 3b–c). Fraction containing the purified complex was with crosslinked with 0.05% w/v glutaraldehyde on ice for 10 min before quenching with 16 mM aspartate and 4 mM lysine. Cryo-EM samples were concentrated using a 50 kDa MWCO Amicon Ultra Centrifugal Filter (Merck).

For grid preparation, R2/2 UltrAuFoil grids (Quantifoil) were glow-discharged for 100 s. 2.5 µL of the sample was applied to both sides of the grids and incubated for approximately 5 s at 4 °C and 100% humidity. The grids were blotted with a blot force of 5 for 3 s before plunging into liquid ethan using a Vitrobot Mark IV (Thermo Fisher).

## Cryo-EM data collection

Cryo-EM data were acquired at a nominal magnification of 81,000×, corresponding to a calibrated pixel size of 1.05 Å/pixel, using a K3 direct electron detector (Gatan) on a Titan Krios transmission electron microscope (Thermo Fisher Scientific) operated at 300 kV. Images were collected in EFTEM mode using a Quantum LS energy filter with a slit width of 20 eV. Images were collected in electron counting mode with an applied defocus range of 0.5 to 2.0 µm. SerialEM software was used for automated data acquisition[71]. All pre-processing of the collected movies (motion correction, dose weighting, CTF estimation and particle picking) were performed using Warp[72].

We collected approximately 70,000 micrographs with a dose rate of 19.46 e⁻/pixel/s for 2.27 s resulting in a total dose of 39.74 e⁻/Å² that was fractionated into 40 movie frames. Micrographs with bad CTF fits in Warp were excluded from further processing. We extracted approximately 10.1 million particles with a box size of 512 pixels (binned 4x) to a pixel size of 4.2 Å/pixel using RELION 5.0[73]. These particles were subjected to iterative rounds of 2D classification, and heterogenous refinement in CryoSPARC[74] using initial models generated by ab initio reconstruction. Classes with good particles of RNA polymerase were re-extracted and re-centered in RELION 5.0 (binned 2x, pixel size 2.1 Å/pixel). After further rounds of 2D and 3D classification in CryoSPARC and RELION remaining particles were re-extracted both binned 2x (pixel size 2.1 Å/pixel) and unbinned (pixel size 1.05 Å/pixel). Unbinned particles (approximately 1.5 million) were subjected to iterative rounds of CTF refinement and Bayesian polishing resulting in a 2.5 Å reconstruction, encompassing the activated elongation complex. Further classification for the presence of elongation factors, including ELOF1 and IWS1 were performed on the binned 2x particles in RELION (without angular sampling unless indicated otherwise) using local masks (Supplementary Fig. 4). Classification of the downstream nucleosome was performed using 2D classification, heterogenous refinement and non-uniform refinement in CryoSPARC. Following a single round of 3D classification, approximately 1800 particles were common between selected particles of individual classes and were selected from the unbinned consensus refinement, resulting in a 4.07 Å reconstruction after local refinement in CryoSPARC (with a mask around Pol II).

To further improve resolution of the classified elongation factors, signal subtraction with centering was performed on unbinned particles using local masks[75]. 3D classification in RELION was performed, utilizing Blush as indicated[76] (Supplementary Fig. 5). After sorting, selected particles subjected to local refinement in CryoSPARC. To demonstrate the local position on Pol II, various overlapping subsets of elongation factors were selected from the consensus refinement and refined with a local mask around Pol II in CryoSPARC.

To highlight variation in the downstream DNA position, particles containing ELOF1 (binned 2x) were subjected to local classification without angular sampling. Only 1 class contained density for the C-terminal ECS domain of IWS1. Particles from each class were selected from the unbinned consensus refinement and an additional refinement performed in RELION with a Pol II mask.

## Model building

To build a model of the activated elongation complex with IWS1 and ELOF1, focused refined Maps 4–10 were fitted into the density of Maps 10–12 and used to guide position of PDB 6TED with RTF1 removed[61]. Cryo-EM maps 8 and 9 that incorporate newly identified regions of IWS1, as well as the interaction with ELOF, were of sufficiently high-resolution to unambiguously assign the register to both of these proteins. To aid model building, the AlphaFold 3 prediction of Pol II, ELOF1 and IWS1 was rigid body docked into Map 9 and incorporated into the overall model. The PAF1 / LEO1 heterodimer of 6TED was replaced with the AlphaFold 3 predication that was rigid body docked into Map 4 and showed good agreement with our observed density. Map 5 was used to place the Alphafold 3 prediction of the structured domain of CDC73 with local adjustment of the SPT6 tSH2 domain. Once placed, residues were locally fitted in ISOLDE[77] whilst maintaining secondary structure, torsion and distance restraints. Final data processing and model validation statistics are available in Supplementary Table 2.

## IWS1 sequence alignments

The sequence alignment shown in Supplementary Fig. 9 of was generated using ClustalOmega[78] and visualized using Jalview[79].

## In vitro methylation assay

In vitro methylation assays in the presence or absence of transcription and IWS1 were performed on a chromatinized DNA template containing 4x Widom 601 positioned nucleosomes separated by 30 bp of linker DNA. The DNA template sequence is detailed in Supplementary Table 1. The initial template was generated by restriction enzyme digest from plasmid DNA. For transcription, extension of an RNA primer, complimentary to a TspRI generated single stranded DNA overhang on the 5' end of the template, was performed as previously described[24,69]. The concentrations listed refer to the final concentrations used in the assay. RNA primer (150 nM) was incubated with chromatinized template (60 nM) and *S. scrofa* Pol II (150 nM) on ice for 10 min. EC* components DSIF (225 nM), SPT6 (225 nM), RTF1 (225 nM), and PAF1 complex (225 nM) were added along with P-TEFb (225 nM), FACT (300 nM), AKT3 (400 nM) and ATP (1 mM). Buffer, IWS1 (450 nM) or SETD2 (600 nM) were added as described in Supplementary Fig. 11h–i and incubated for 5 min at 30 °C. Transcription was initiated by the addition of CTP, GTP, UTP (0.5 mM each), TFIIS (90 nM) and SAM (4 μM). The final reaction conditions in 15 μL were 100 mM NaCl, 20 mM Na-HEPES, pH 7.4 at 20 °C, 5 mM MgCl$_2$, 1 mM DTT, and 4% glycerol. After 30 min incubated at 30 °C, 12 μL of sample was mixed with 4 μL of LDS loading dye and subjected to PAGE (12% Bis-Tris) and subsequently transferred onto nitrocellulose membranes (GE Healthcare Life Sciences) for Western blotting using an antibody against histone H3K36me3 (Active Motif, REF: 61021), that was carried out following standard protocols.

## Chromatin proteomics

Chromatin proteomics was performed in four biological replicates. For each replicate, 10 million cells were treated with DMSO or dTAG7 for 1 h or 4 h, followed with the fractionation protocol[67]. MNase digestion was stopped with 100 μl 2X SDS buffer (100 mM Tris-HCl, pH 7.4, 4% SDS) to solubilize proteins. The sample was boiled at 99 °C for 5 min, then centrifuged at 10,000 g for 10 min to pellet insoluble material. The protein concentration was measured using Pierce BCA Protein Assay Kit (Thermo Fisher Scientific). 200 μg of proteins was reduced and alkylated with 10 mM dithiothreitol (DTT) at 37 °C for 30 min and 40 mM IAM at 37 °C for 30 min in the dark. A mixture of Sera-Mag SpeedBeads (GE Healthcare) was rinsed twice with water and then added to protein lysates at the working ratio of 10:1 (w/w, beads to proteins). After adding acetonitrile (ACN) to a final percentage of 50% (v/v), the beads and proteins were incubated for 10 min at room temperature off the rack, followed by resting on magnetic rack for 2 min to remove the supernatant. The beads were washed three times with 90% (v/v) ACN and once with 100% (v/v) ACN, resuspended in 50 mM triethylammonium bicarbonate (TEAB) with 1 mM MgCl$_2$. 1 μL benzonase was added and incubated at 37 °C for 3 h, followed by the incubation with sequencing grade modified trypsin (1:20 of enzyme-to-protein ratio) at 37 °C overnight in a ThermoMixer with mixing at 1000 rpm. The resulting peptides in the supernatant were collected, completely dried in a SpeedVac and stored at −80 °C until further use.

LC-MS/MS analysis was performed on an Orbitrap Exploris 480 Mass Spectrometer (Thermo Fisher Scientific) coupled to a Dionex UltiMate 3000 UHPLC system (Thermo Fisher Scientific). Peptides were separated on an in-house packed analytical column (30 cm length; ReproSil-Pur 120 Å, 3 μm pore size, C18-AQ; 75 μm inner diameter) with a gradient of 5–9% solvent B (0.08% (v/v) formic acid, 80% (v/v) acetonitrile) in 3 min, 9–34% B in 92 min, 34–45% B in 11 min and 90% B in 6 min at a flow rate of 300 nl/min. The MS instrument settings in data-independent acquisition mode are described briefly. A total of 40 variable windows covering a mass range of 350–1650 m/z were used. The resolution was set to 120,000 for MS1 and 30,000 for MS2. The AGC target mode was set to 300% for MS1 and 1000% for MS2, with a maximum injection time of 20 ms in MS1 and 55 ms in MS2. NCE was set to 30%.

## TT-seq

TT-seq was performed as previously described[20,51], utilizing two biological replicates. For each replicate, 30 million cells were treated with dTAG7 or DMSO solvent for 1 h, followed by 5 min incubation with 500 μM 4-thiouridine (4sU) to label the RNA. Subsequent to labeling, cells were lysed using Qiazol (QIAGEN), and the lysates were spiked with three 4sU-labeled and three unlabeled synthetic spike-in RNAs. The extraction of total RNA was then carried out using the standard Qiazol-chloroform extraction method. Post-extraction, the RNA was fragmented for 10 s using an S220 Focused-ultrasonicator (Covaris). This was followed by the biotinylation of 4sU-incorporated RNA fragments. For each sample, biotinylation was executed in two separate reactions, each containing 150 μg of fragmented RNA, 10 mM Tris-HCl (pH 7.5), 1 mM EDTA (pH 8.0), 200 μg/mL HPDP-biotin, and 40% DMSO. The reactions were then combined post-purification using the μMACS Streptavidin Kit (Miltenyi). The 4sU-labeled RNA was eluted using 100 mM DTT and further purified with the miRNeasy Micro kit (Qiagen). 100 ng of the purified, 4sU-labeled RNA was used for NGS library preparation using the Stranded Total RNA Prep with Ribo-Zero Plus kit (Illumina), according to the manufacturer's instructions with a 5-min RNA fragmentation step and seven cycles of PCR amplification. Finally, the quality of the libraries was assessed using a Fragment Analyzer. The libraries were then pooled and sequenced paired-end with a 75-cycle sequencing run on a Nextseq550 (Illumina) platform.

## mNET -seq

mNET-seq was performed as previously described[53,54,56,80,81], utilizing two biological replicates with slight modifications. Following dTAG7or DMSO treatment, 100 million cells were fractionated as described previously[67]. To protect against degradation, all fractionation buffers were supplemented with 1x protease and phosphatase inhibitors (Sigma-Aldrich). Following fractionation, the isolated chromatin was subjected to a 2-min digestion at 37 °C and 1400 rpm using micrococcal nuclease (MNase, NEB). The reaction was halted by adding 25 mM EGTA. The resultant soluble chromatin fraction was then diluted 8-fold with IP buffer (50 mM Tris-HCl pH 7.5, 150 mM NaCl, 0.05% NP-40, 1% Empigen, 1x protease and phosphatase inhibitors (Sigma-Aldrich)). For IP, 30 μg of Pol II antibody (Diagenode, REF: C15200004) was pre-coupled overnight with Dynabeads M-280 Sheep Anti-Mouse IgG (Invitrogen) and added to each diluted sample. The mixtures were incubated for 1 h at 4 °C. Following IP, the beads were washed seven times with IP buffer and once with PNKT buffer (1x T4 PNK buffer (NEB), 0.1% Tween-20). The 5' ends of RNA were phosphorylated using T4 Polynucleotide Kinase (PNK, NEB) for 10 min at 37 °C and 800 rpm. After a final wash with IP buffer, the beads were resuspended in TRIzol reagent (Invitrogen), and 2.5 ng of *S. cerevisiae* spike-in RNA, prepared as per[56], was added to each sample. RNA was extracted from the beads and precipitated overnight with ethanol and GlycoBlue co-precipitant (Invitrogen) at −20 °C. The RNA pellets were then dissolved in buffer containing 7 M urea, and size selection was carried by size separation in a 6% polyacrylamide gel containing 7 M urea and excision of 25–110 nt long RNA fragments. The recovered RNA was precipitated again with GlycoBlue and ethanol at −20 °C overnight. For library preparation, 28.95 ng of RNA from each replicate was used with the TruSeq Small RNA Library Prep Kit (Illumina). The quality of the libraries was assessed using TapeStation (Agilent), followed by paired-end sequencing with 42 cycles on the NextSeq550 (Illumina).

## ChIP-seq

ChIP-seq was performed as previously described[28], utilizing two biological replicates. For each replicate, 30 million cells were treated with dTAG7 or DMSO-solvent for 1 h or 4 h and crosslinked using 1% formaldehyde (Thermo Fisher Scientific) and chromatin was extracted. The crosslinked chromatin was then sheared using an S220 Focused-ultrasonicator (Covaris) for 18 min. Subsequently, 100 μg of the

sheared chromatin was incubated with one of the antibodies pre-coupled to Dynabeads Protein G (Invitrogen): 20 μL Rpb1 NTD (Cell Signaling Technology, REF: D8L4Y) or 20 μL Histone H3K36me3 (Active Motif, REF: 61021) was added to each sample. Following IP, the DNA was purified from the complexes, and library preparation was conducted using the NEBNext Ultra II DNA Library Prep Kit for Illumina (NEB). The prepared libraries were then sequenced paired-end using 43 cycles on the NextSeq550 (Illumina) platform.

### Computational data analysis

**Chromatin proteomics data processing.** The MS raw files were imported into DIA-NN (version 1.8.1)[82], UniProt human protein database with 20,598 entries (released on March 27, 2024) was used to generate the spectrum library. Mass ranges were set appropriately for the search space (MS1: 300 m/z to 180 m/z; MS2: 200 m/z to 1800 m/z). Cleavages were specified at lysine (K) and arginine (R) residues. Maximum number of missed cleavages set to 1. N-terminal methionine excision was enabled. Carbamidomethylation of cysteine residues was enabled as a fixed modification, oxidation of methionine residues was set as a variable modification. Protein and precursor FDR were set to 1%.

Data visualization was performed on ggplot2[83] on RStudio, R version 4.3.1. Center-median normalized log2 transformed intensity were used as protein abundance.

**High-throughput sequencing data preprocessing.** All sequencing data were initially processed using Illumina's demultiplexing tools. We assessed the quality of the sequencing reads with FastQC, to ensure data integrity and readiness for downstream analyses.

TT-seq reads were aligned to the human reference genome (GRCh38) that had been augmented with synthetic RNA spike-in sequences. We utilized STAR version 2.6.0[84] for alignment with the following parameters: --runThreadN 24 --readFilesCommand zcat --outFilterType BySJout --outFilterMultimapNmax 1 --alignSJoverhangMin 8 --alignSJDBoverhangMin 1 --outFilterMismatchNmax 999 --outFilterMismatchNoverLmax 0.02 --alignIntronMin 20 --alignIntronMax 1000000 --alignMatesGapMax 1000000.

For mNET-seq, the alignment was performed against a composite reference consisting of the human GRCh38 genome and the *S. cerevisiae* SacCer3 genome, using STAR version 2.6.0[84] with parameters: --runThreadN 24 --readFilesCommand zcat --outFilterType BySJout --outFilterMultimapNmax 1 --outFilterMismatchNoverLmax 0.02 --outFilterMatchNmin 16 --outFilterScoreMinOverLread 0 --outFilterMatchNminOverLread 0 --alignIntronMax 500000.

For TT-seq and mNET-seq, the quantifications of reads aligning to the spike-in sequences were used to calculate normalization factors, ensuring accurate comparison of expression levels across samples as in ref. 28. Spike-in read counts were assessed by summarizeOverlaps function (Bioconductor package "IRanges") and normalization factors were calculated with DESeq2[85].

ChIP-seq reads were aligned to a hybrid reference combining the human GRCh38 genome, employing Bowtie2 version 2.3.4.1[86] with options: -p 24 --no-discordant --no-mixed --very-sensitive. Picard tools (http://broadinstitute.github.io/picard) were used to remove the duplicates. ChIP-seq data was normalized to the coverage at non-transcribed regions[57]. ChIP-seq counts at non-transcribed regions were assessed using summarizeOverlaps function (Bioconductor package "IRanges") and normalization factors were calculated with DESeq2[85].

Utilizing the RefSeq reference annotation[87], we refined our gene set by excluding overlapping genes and selecting those with significant expression based on median RPK values from TT-seq data. A threshold of RPK > 10 was used to define expressed genes, resulting in a curated set of 10,147 genes for downstream analysis.

All downstream analyses were conducted in RStudio, using R version 4.3.1. The analysis relied on packages sourced from the

Bioconductor repository[88,89] and the Tidyverse suite[90], with graphical representations generated through the ggplot2 package[83].

**Differential gene expression analyses.** For TT-seq, mNET-seq and ChIP-seq samples, counts per gene were generated using the summarizeOverlaps function (Bioconductor package "IRanges") from GRanges of the last incorporated base. This function was configured with specific parameters to ensure accurate gene counting: ignore.strand = FALSE, singleEnd = FALSE, mode = "IntersectionNotEmpty". Differential gene expression analysis was performed with the DESeq2 package[85]. We incorporated the normalization factors derived from spike-in counts as sizeFactor parameters within DESeq2. MA plot was generated using the plotMA function from DESeq2. For boxplots Figs. 1c, 2a, 5e, g, i; Supplementary Figs. 1e, 2b, c, f, 11d, g, j, n counts were taken from gene bodies (TSS to TES).

**Intron Retention (IR) ratios.** For checking the changes in splicing efficiencies, we used IRFinder tool[91], version 1.3.0, on TT-seq fastq files with the following parameters: --outFilterMismatchNmax 999 --outFilterMismatchNoverLmax 0.02.

**Estimation of elongation velocity.** We determined elongation velocity using the ratio of TT-seq to mNET-seq data, which reflects the relationship between RNA synthesis and Pol II occupancy, following the approach by ref. 20. Described here in brief, the GRCh38 genome was segmented into 50 bp bins, and for each, we calculated the ratio of normalized TT-seq midpoint coverage to mNET-seq coverage. These ratios, representing elongation velocities for both DMSO- and dTAG7-treated control samples, were visualized in heatmap (Fig. 2d). For the boxplots (Fig. 2e), we calculated the ratio of normalized TT-seq counts to normalized mNET-seq counts in the gene body without first 1 kb.

**Metagene profiles and heatmaps.** We generated metagene profiles and heatmaps for TT-seq, mNET-seq and ChIP-seq data following the procedures outlined in refs. 20,28, with slight modifications. For each assay, normalized coverages from DMSO/dTAG7-treated controls were combined by summing up per nucleotide values across replicates. To facilitate log transformation, we added a pseudo-count of 1 to each position, calculated the log2-transformed means per base pair/bin, and determined mean and its 95% confidence intervals using bootstrap analysis with 10,000 iterations. For H3K36me3 ChIP-seq, heatmaps were produced similarly after tiling the human genome into 500 bp bins; additionally, to minimize the influence of adjacent genes, regions outside the gene bodies were masked.

**Schematic representations.** Schematics in Fig. 1a, Supplementary Fig. 11h were created with Biorender.com.

### Reporting summary

Further information on research design is available in the Nature Portfolio Reporting Summary linked to this article.

## Data availability

NGS datasets generated in this study have been deposited in NCBI's Gene Expression Omnibus database (GEO) and are accessible through GEO Series accession numbers GSE276547, GSE276549, GSE276550. The mass spectrometry proteomics data have been deposited to the ProteomeXchange Consortium via the PRIDE[92] partner repository with the dataset identifier PXD056248 and PXD066658. The cryo-EM reconstructions corresponding to MAPS 1–12 have been deposited in the Electron Microscopy Data Bank under accession codes EMD-54196, EMD-54197, EMD-54201, EMD-54202, EMD-54203, EMD-54204, EMD-54205, EMD-54206, EMD-54207, EMD-54208, EMD-54209, EMD-54210, EMD-54211, and EMD-54212. The atomic model of the activated RNA polymerase II elongation complex bound to IWS1 and ELOF1 has been

deposited in the Protein Data Bank under accession code 9RTT. Source data are provided with this paper.

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

## Acknowledgements
We thank Sara Ahrari for the help with ChIP-seq experiments. We thank Michelle Kobold and Monika Raabe for assistance in mass spectrometry. We thank Christian Dienemann and Ulrich Steuerwald for maintaining the cryo-EM infrastructure. We are grateful to Michael Lidschreiber, Christian Dienemann, Taras Velychko and Kseniia Lysakovskaia for their feedback on the manuscript. We thank all past and present members of the Cramer Lab for many productive scientific discussions. A.Z. was supported by the International Max Planck Research School for Genome Science, which is a part of the Göttingen Graduate School for Neurosciences, Biophysics, and Molecular Biosciences. P.C. was supported by the Max Planck Society.

## Author contributions
A.Z., J.L.W., K.Ž., and P.C. conceived and planned the study; A.Z. designed and performed in vivo experiments, bioinformatic analysis, and interpreted data; J.L.W. designed and performed cryo-EM experiments and interpreted data; J.L.W., M.O., and M.B. purified proteins and nucleic acids, designed and performed in vitro transcription and methylation experiments with U.N.; K.C.M., P.R. generated the dTAG-IWS1 cell line; Y.Y. analyzed MS experiments under H.U. supervision.; P.C. acquired funding; K.Ž. and P.C. supervised the study; A.Z. and J.L.W. wrote the original manuscript draft; K.Ž. and P.C. reviewed and edited the manuscript, with input from all authors.

## Funding

## Competing interests
The authors declare no competing interests.
