## [Transparent Peer Review file · Nature Communications]

IWS1 positions downstream DNA to globally stimulate Pol II elongation

Corresponding Author: Dr Kristina Zumer

A version of this paper was originally rejected for publication by Nature Communications, however that decision was reconsidered after appeal by the authors.

Version 0:

Reviewer comments:

Reviewer #1

(Remarks to the Author)

IWS1 is a conserved transcription elongation factor that is part of the RNA polymerase II elongation complex. Initially discovered and studied in yeast, IWS1 also has essential roles in mammalian cells. In this manuscript, the authors study the requirement for IWS1 for transcription and for H3K36me3 by studying a cell line before and after IWS1 depletion. Their results clearly show a decreased elongation rate over the majority of genes after IWS1 depletion. They also observe decreased levels of H3K36me3, but based on SetD2 ChIP-seq experiments, they conclude that this change is a consequence of the reduced rate of transcription. The results are novel and will be of interest to the transcription community. There are several issues to address regarding the experiments and the writing, in particular many cases where the literature citations need to be improved. The comments are presented in the order they arose while reading the manuscript.

1. line 21 - I suggest changing the first two sentences as follows: in the first sentence, replace "Metazoan" with "Eukaryotic" and in the second sentence, change the beginning to "In humans, its misregulation..."
2. line 26 - I'm not sure why reference 10 is cited for a review of elongation factors. I suggest including a recent review by Lucas Farnung - PMID: 39476950.
3. line 28 - In addition to the Vos et al. paper, reference 12 (Wada et al.) should be cited for the negative role of DSIF.
4. line 34 - There are newer reviews that should be cited for both Spt6 (Miller et al. 2023, PMID: 37481442) and PAF (Francette et al. 2021, PMID: 39476950).
5. line 35 - Remove "the" from the beginning of the line.
6. line 38 - There is a review of FACT that might be better to cite instead of or in addition to the papers already cited - Formosa and Winston 2020, PMID: 33104782).
7. line 42 - IWS1 was initially discovered by three independent studies in yeast, all of which linked it to transcription and transcription-related processes: (Krogan et al. 2002, PMID: 12242279; Fischbeck et al. 2002, PMID: 12524336; and Lindstrom et al. 2003, PMID: 12556496). These should be cited in addition to, or in place of those cited.
8. lines 41-56 - This paragraph introduces IWS1 but fails to include the interaction upon which it was named, its interaction with SPT6. This is an important part of the background. The relevant papers are McDonald et al. 2010, PMID: 21094070 and Diebold et al. 2010, PMID: 21057455.
9. line 50 - What is an in vitro histone chaperone activity? Is that different from histone binding?
10. The authors neglect to properly cite and discuss a major paper about IWS1 (reference 54 in their citation list). This paper is cited briefly, but it has a lot of data about IWS1 interactions with other proteins, as well as transcriptional analysis of an

IWS1 mutant (not depletion). This is very relevant to the current manuscript. Please discuss and cite in the introduction and elsewhere where appropriate, such as by comparing their PRO-seq results to the mNET-seq results.

11. Was anything done to test whether the N-terminal dTAG affects IWS1 function, such as comparing TT-seq of an untagged cell line to the undepleted dTAG-IWS1 cell line?

12. Figure 1 and Extended Data Fig. 1a – I suggest adding Extended Figure 1b to Figure 1 as it provides more information than the metagene in Figure 1d.

13. It would be helpful to readers to compare the TT-seq result for IWS1 depletion to the TT-seq results for the depletion of other elongation factors that have been done by this lab, such as Ssrp1, Spt6, Rtf1. How do they compare in terms of % of genes and degree of change?

14. Figure 2 – There is a difference between the mNET-seq and the CHIP-seq results that is not commented upon: an obvious 5' accumulation of Pol II observed by mNET-seq that is not observed by CHIP-seq. The 5' effect is also evident in 2d. Please mention and discuss.

15. Figure 3c,d – What is the barrier index and how is it calculated? It would be clearer to call this the short transcript and measure its level similar to what is done in 3d. For 3d, please explain in methods how normalized intensity was measured. Normalized to what?

16. line 142 – What happens to SETD2 recruitment after 4 hours of IWS1 depletion?

17. Figure 3g – SETD2 and H3K36me3 are usually recruited across transcribed sequences. Please explain how “peaks” are defined.

18. lines 151-153, Figure 3h – There are a few points regarding this result and the way it is described in the text. First, the experiment in reference 47 consisted of analysis of a single promoter under a certain set of growth conditions, a very different experiment than the proteomic analysis presented here, particularly since they were looking at a promoter region, a location where Spt6 and Iws1 are not normally found. Second, reference 29 found that there is modest decrease in Spt6 recruitment after IWS1 (Spt1) depletion. Finally, the writing will be clearer if the sentence ending on line 153 added “after IWS1 depletion” to the end of the sentence.

19. In methods, it is stated that cells were treated for one hour for TT-seq, but this information is not provided for mNET-seq or CHIP-seq. Also, the number of replicates is not stated for CHIP-seq.

20. Please provide information on the reproducibility between the replicates for the TT-seq, mNET-seq, and CHIP-seq. Also, please clarify what is presented in the TT-seq, mNET-seq, and CHIP-seq figures. Are we looking at one of the two replicates, or merged datasets?

21. Figure 3j – Please tell us the number of replicates for this experiment and include quantitative analysis of the results.

22. lines 168-169 – If the change in H3K36me3 is a result of altered transcription than this would be evident in a scatterplot of the change in TT-seq vs. the change in H3K36me3. This information should be presented.

23. line 421 – change “initially” to “initial”

Reviewer #2

(Remarks to the Author)

This study demonstrates that IWS1 plays a direct role in transcription elongation by supporting RNA polymerase II (RNAP2) activity. Its rapid depletion reduces elongation velocity and has secondary effects on H3K36me3 levels without affecting the recruitment of the H3K36me3 writer, SETD2. IWS1 is an exciting target, known to directly interact with and regulate RNAP2-mediated transcription. While the authors' approach is innovative and holds significant potential to uncover key dependencies in RNAP2 regulation, the manuscript does not meet the standards I would expect from this lab. Specifically, the genome-wide data analyses are preliminary and lack robust statistical support. Additionally, mechanistic insights into how IWS1 regulates RNAP2 activity and impacts H3K36me3 deposition or turnover are limited. Addressing this issue would provide the novelty currently lacking. Several major points are outlined below:

Major Comments:

1. The authors did not provide a pre-release token for their GEO data, making their genome-wide data inaccessible to reviewers. As a result, it is not possible to assess the quality of the genomic data and evaluate their data interpretation.

2. The statement in the abstract that “whether IWS1 has a role in RNA polymerase (Pol) II function remains unclear”

overlooks previous studies that clearly demonstrate IWS1's significant role in RNAP2 regulation (PMIDs: 32941642, 34822292, 17234882, 29300974, 38559248, 33888559, and others).

3. It is critical that the authors validate the targeted protein degradation of IWS1 using an IWS1-specific antibody. This is particularly important because the study relies on a CRISPR/Cas9 dTAG knock-in cell line, and it is crucial to confirm that no endogenous (untagged) IWS1 remains after the knock-in. Currently, the depletion of the IWS1-HA-HA-dTAG protein product is assessed using an HA antibody, which detects only the tagged IWS1 and cannot rule out the presence of endogenous IWS1.

4. This paper presents itself as the first to reveal the role of IWS1 in transcription but does not extend beyond what has already been shown. For example, prior studies demonstrated an acute role for the yeast IWS1 homolog Spn1 in normal transcription, splicing of ribosomal protein transcripts, and proper localization of H3K36me3 along coding genes (PMID: 32941642). The rapid degradation of SPT6, an interaction partner of RNAP2, highlighted its crucial role in transcription termination (PMID: 34233157). Additionally, another study showed that mutations in IWS1 disrupting its interaction with H3K36me3 readers result in defects in releasing paused RNAP2 into productive elongation (PMID: 34822292). Does IWS1 degradation similarly affect transcription start site selection (TSS), transcription termination, or the release of promoter-proximal paused RNAP2 into elongation? The authors have already generated datasets that could address these questions.

5. The authors observe that while global SETD2 occupancy remains unchanged, H3K36me3 levels are reduced after IWS1 degradation. The authors conclude that the primary effect of IWS1 depletion is on transcription elongation, which subsequently has a secondary effect on H3K36me3 deposition. This observation is intriguing; however, it remains unclear how the observed reduction in H3K36me3 levels occurs within such a short timeframe. Turnover of H3K36me3 after SETD2 inhibition is a lengthy process, typically taking hours to days. How does the transcriptional disruption caused by IWS1 depletion lead to accelerated turnover of this histone mark? What mechanism do the authors propose to explain this rapid reduction in H3K36me3?

6. The authors claim that H3K36me3 is a secondary effect of transcriptional deregulation. Is there a relationship between the observed transcriptional downregulation and the decrease in H3K36me3 levels at individual gene bodies? The authors need to show this to support their claim.

7. The authors should provide a more comprehensive analysis of their ChIP-seq data beyond the box plots currently presented. Specifically, they should provide separate heatmaps for each replicate to clearly show the loss of H3K36me3 and the retention of SETD2 across the relevant genomic regions. Additionally, example browser tracks should be included to visually demonstrate these changes. Without these additional analyses, it will be difficult for readers to fully trust the conclusions.

8. The authors do not provide statistical measures to support many of their claims. Of particular concern is the main claim that rapid depletion of IWS1 leads to an increase in RNAP2 occupancy at genes and a decrease in elongation velocity. This central finding is not appropriately supported by statistical analysis. This issue should be addressed throughout the manuscript to ensure that all the authors' claims are substantiated.

9. Figure 2A must include control data to compare the RNAP2 occupancy before IWS1 degradation. As it currently stands, the figure is incomplete.

Reviewer #3

(Remarks to the Author)

This is a well done manuscript which shows that the mammalian IWS1 protein stimulates transcription in vivo and in vitro. The experiments are well done and the conclusions are solid. It is a short report and conclusive.

The difficulty I have with the manuscript is that it does not provide significant novel information beyond what is known previously about IWS1 and its yeast homolog. It lacks mechanistic insight other than to suggest its activity is not mediated through recruitment or activity of SETD2. While these confirming results will be of interest to the elongation field it does not provide enough novel insights for publication in Nature Communications. The current work could serve as an important part of a longer paper providing additional data addressing mechanisms or it could be published in a more specialized journal.

Version 1:

Reviewer comments:

Reviewer #1

(Remarks to the Author)

The authors have done a nice job of responding to the reviewers' comments. The addition of the structural studies adds interesting new results. I think that the manuscript now merits publication.

Reviewer #2

(Remarks to the Author)

The authors have addressed some, but not all, concerns from the initial review. Several major issues remain, primarily because key analyses were conducted for the reviewers but not incorporated into the manuscript. The addition of the structural model raises further concerns, particularly due to its reliance on unvalidated AlphaFold predictions and the weak integration of the new data into the overall narrative. Please see detailed comments below:

1) The authors performed a substantial number of analyses to address the reviewers' concerns, including those raised by me and Reviewer 1. However, many of these analyses were shared only as rebuttal figures in the response to reviewers and were not incorporated into the manuscript. This approach does not adequately resolve the issues raised. While not every analysis needs to be included, many are directly relevant and should be presented in the manuscript, as readers are likely to raise similar questions.

2) The authors present a new structural analysis extending existing models by identifying additional interfaces of IWS1, including an "elongation complex scaffold domain" proposed to stimulate RNA Polymerase II. Since IWS1's disordered regions are established in the literature as interaction scaffold for multiple elongation factors, the novelty of this interface relies on the identification and structural analysis of the interfaces. However, this disordered domain is modeled by fitting an AlphaFold predictions into unassigned cryo-EM density without experimental validation—neither mutagenesis nor structural mapping (e.g., X-ray crystallography, NMR, XL-MS) has been performed. While the authors demonstrate that deleting the entire domain impairs Pol II stimulation, the individual contacts remain unvalidated. Experimental confirmation of these predicted interfaces is essential to substantiate the structural claims. Without such validation, the proposed scaffold domain remains speculative and should not be presented as a confirmed structural feature.

3) The authors have added new structural data to enhance the manuscript; however, ELOF1 is not well integrated into the overall narrative. The study is primarily centered on IWS1, but the inclusion of ELOF1 disrupts the focus and weakens the framing. Does ELOF1 have a unique functional relationship with IWS1 that distinguishes it from other elongation factors? If so, this should be clearly articulated. Why was ELOF1 chosen as a focal point over other well-characterized elongation factors that also interact with IWS1 such as TFIIIS, SPT6, or SPT5?

4) The authors refer to both the IWS1 "core domain" and the "TND domain" throughout the manuscript. However, IWS1 has only one folded domain—commonly known as the TND—which includes an additional HEAT repeat. While a recent yeast transcription complex study used the term "core domain," the majority of the literature refers to this region as the TND, sometimes specifying the HEAT repeat. Switching between two different terms without clear explanation is misleading and may imply the presence of two separate domains. The authors should clarify this terminology and consider adopting the more widely accepted nomenclature, particularly since the manuscript includes structural data and the terms "TND" and "HEAT repeat" accurately reflect the domain's architecture.

5) The explanation of Figure 2A does not address my concern. Presenting a bar graph with a single column showing fold change, p-value, and standard deviation—without including the reference or baseline group—is problematic. Since fold change is inherently comparative, omitting the baseline obscures essential context and limits the interpretability of the results. This is particularly important in genomics, where subtle differences can yield statistically significant p-values without reflecting biologically meaningful effects.

6) Minor issue: The authors clearly state that IWS1 is N-terminally tagged with dTAG; however, the labeling throughout the manuscript—especially in the figures—is potentially confusing. For instance, Figure 1a lacks sufficient resolution to show the orientation of the IWS1 gene, which may give the impression that the tag is C-terminal. Additionally, the notation "IWS1-dTAG(HA)" suggests a C-terminal tag. I recommend clearer and more consistent labeling to prevent misinterpretation.

Reviewer #3

(Remarks to the Author)

my concern with the original manuscript was the limit of novel results that was reported. The new structural and biochemical experiments included in the revised manuscript extend the story and provide additional novel mechanistic insights. The revised manuscript is suitable for publication.

Version 2:

Reviewer comments:

Reviewer #2

(Remarks to the Author)

The authors have addressed my concerns, and I appreciate the effort they put into the revisions. The revised manuscript is strong, and I recommend it for publication.

Point-to-point response - reviewers' comments:

Reviewer #1 (Remarks to the Author):

IWS1 is a conserved transcription elongation factor that is part of the RNA polymerase II elongation complex. Initially discovered and studied in yeast, IWS1 also has essential roles in mammalian cells. In this manuscript, the authors study the requirement for IWS1 for transcription and for H3K36me3 by studying a cell line before and after IWS1 depletion. Their results clearly show a decreased elongation rate over the majority of genes after IWS1 depletion. They also observe decreased levels of H3K36me3, but based on SetD2 ChIP-seq experiments, they conclude that this change is a consequence of the reduced rate of transcription. The results are novel and will be of interest to the transcription community. There are several issues to address regarding the experiments and the writing, in particular many cases where the literature citations need to be improved. The comments are presented in the order they arose while reading the manuscript.

We thank the reviewer for the evaluation and detailed review of our manuscript, with helpful specific suggestions how to improve it.

We have added new results, which include a structure of the activated elongation complex with IWS1 and ELOF1, additional biochemical assays and new cell-based experiments, which led the reorganization of the entire manuscript with many changes in the manuscript text and figures. Importantly, our conclusions that IWS1 directly stimulates Pol II activity and that the decrease in H3K36me3 are a secondary effect of perturbed elongation remain unchanged, and we now provide molecular insight into the role of IWS1 in elongation.

Points 1-8 are very helpful and improved the introduction section. They were addressed with minor text edits, additional references and additional elaboration of terminology.

1. line 21 - I suggest changing the first two sentences as follows: in the first sentence, replace "Metazoan" with "Eukaryotic" and in the second sentence, change the beginning to "In humans, its misregulation..."
2. line 26 – I'm not sure why reference 10 is cited for a review of elongation factors. I suggest including a recent review by Lucas Farnung - PMID: 39476950.
3. line 28 – In addition to the Vos et al. paper, reference 12 (Wada et al.) should be cited for the negative role of DSIF.
4. line 34 – There are newer reviews that should be cited for both Spt6 (Miller et al. 2023, PMID: 37481442) and PAF (Francette et al. 2021, PMID: 39476950).
5. line 35 – Remove "the" from the beginning of the line.
6. line 38 – There is a review of FACT that might be better to cite instead of or in addition to the papers already cited – Formosa and Winston 2020, PMID: 33104782).
7. line 42 – IWS1 was initially discovered by three independent studies in yeast, all of which linked it to transcription and transcription-related processes: (Krogan et al. 2002, PMID: 12242279; Fischbeck et al. 2002, PMID: 12524336; and Lindstrom et al. 2003, PMID: 12556496). These should be cited in addition to, or in place of those cited.
8. lines 41-56 – This paragraph introduces IWS1 but fails to include the interaction upon which it was named, its interaction with SPT6. This is an important part of the background. The relevant papers are McDonald et al. 2010, PMID: 21094070 and Diebold et al. 2010, PMID: 21057455.
9. line 50 – What is an in vitro histone chaperone activity? Is that different from histone binding?

We thank the reviewer for highlighting this point. Reference 44 (Li et al., 2018; PMID: 29300974) performed both H3/H4 deposition and chromatin assembly assays to infer Spn1 to have “weak nucleosome assembly activity” that was described as a “histone chaperone function”. We choose the same termination for consistency.

10. The authors neglect to properly cite and discuss a major paper about IWS1 (reference 54 in their citation list). This paper is cited briefly, but it has a lot of data about IWS1 interactions with other proteins, as well as transcriptional analysis of an IWS1 mutant (not depletion). This is very relevant to the current manuscript. Please discuss and cite in the introduction and elsewhere where appropriate, such as by comparing their PRO-seq results to the mNET-seq results.

We thank the reviewer for highlighting the need to further discuss the relevant findings from Cermakova et al. 2021 (PMID: 34822292), which contains important data on IWS1 interactions and transcriptional analysis of an IWS1 mutant. We have now expanded the Introduction and discussion section to better incorporate the key insights from this study.

To facilitate a direct comparison to our mNET-seq data we have reanalyzed the published PRO-seq data and include plots here. Cermakova et al. (PMID: 34822292) reported that a mutation in the third conserved unstructured motif (TIM3; IWS-M3) increased Pol II accumulation near promoter-proximal regions based on PRO-seq data, whereas we observe a broader accumulation of Pol II across gene bodies following IWS1 depletion with mNET-seq (Fig. 2b). Please note that in the plots we generated from published data we do not observe a global change in accumulation near the TSS for the IWS1 TIM-mutant (IWS1 M3, Rebuttal Fig. 1), which is in line with the figures in the original publication that also only show promoter-proximal accumulation of Pol II for a small subset of genes (n=412). In contrast to this, we observe a genome-wide change in Pol II occupancy in the gene body upon rapid IWS1 depletion (Rebuttal Fig. 2 and Figure 2b). This distinction likely reflects the difference in the indirect role of the TIM3 versus the direct role of the C-terminus of IWS1 in transcription regulation. Our data and conclusions do not exclude a specific role of IWS1 TIM3 in a subset of genes.

Rebuttal Fig. 1. Metagenome analysis of PRO-seq coverage from Cermakova et al. (PMID: 34822292) in non-overlapping genes (n=23036). The scaled metagenome profiles are aligned at the transcript start (TSS) and transcript end site (TES). The y-axis represents the log₂-transformed normalized PRO-seq coverage and the shaded areas around the mean indicate 95% confidence intervals of the mean. The gene region is shaded in gray.

Rebuttal Fig. 2. Metagenome analysis of mNET-seq coverage in non-overlapping genes (n=23036). Representations as in Rebuttal Fig. 1.

11. Was anything done to test whether the N-terminal dTAG affects IWS1 function, such as comparing TT-seq of an untagged cell line to the undepleted dTAG-IWS1 cell line?

We thank the reviewer for this important point. To rule out any impact of the N-terminal dTAG on IWS1 function, we first confirmed by Western blot that parental K562 and dTAG-IWS1 cells express comparable levels of IWS1 (Rebuttal Fig. 3). We re-analyzed TT-seq from K562 with wild-type IWS1 (IWS1-WT) and compared it to IWS1-dTAG \pm dTAG7. To compare these experiments with different amounts of spike-ins, we used sequencing-depth normalization to generate metagene profiles (Rebuttal Fig. 4). We observe that the signal of wild-type and control-treated IWS1-dTAG profiles mostly overlap indicating that elongation is not impacted, whereas the dTAG7-treated IWS1-dTAG exhibits and increased downward-tilt in the profile, which indicates an elongation defect.

The included structural and biochemistry data reveals that the C-terminus of IWS1 is essential for Pol II stimulation. Since the tag lies far upstream of both the SPT6-binding site and our structurally defined Elongation Complex Scaffold with the entire unstructured N-terminal half of the protein between it and the N-terminal tag, the N-terminal dTAG moiety itself unlikely affects IWS1 function.

Rebuttal Fig. 3. Western blots of IWS1 in whole-cell lysates of WT and IWS1-dTAG K562 cells treated with dTAG7 for the indicated time. Histone H3 is a loading control.

Rebuttal Fig. 4. Metagene analysis of TT-seq coverage in non-overlapping genes ($n=23036$) in human K562 cells with IWS1-WT (black) treated with solvent only (DMSO) and IWS1-dTAGed treated with solvent only (orange) (DMSO) or dTAG7 ligand (pink). Representations as in Rebuttal Fig. 1.

12. Figure 1 and Extended Data Fig. 1a – I suggest adding Extended Figure 1b to Figure 1 as it provides more information than the metagene in Figure 1d.

The extended data panel is moved to the main figure and is now Fig. 1d.

13. It would be helpful to readers to compare the TT-seq result for IWS1 depletion to the TT-seq results for the depletion of other elongation factors that have been done by this lab, such as Ssrp1, Spt6, Rtf1. How do they compare in terms of % of genes and degree of change?

We thank the reviewer for this insightful suggestion. To address this point, we utilized differential gene expression analysis results of TT-seq data after RTF1, SPT6 (Zumer et al., 2021; PMID: 34146481) and FACT (Zumer et al., 2024; PMID: 38810649) and directly compared these results with IWS1 depletion. Loss of any elongation factor causes overall reduction of the RNA synthesis (Rebuttal Fig. 5). MA-plots also show that depletion of these elongation factors result in down-regulation of the majority of genes (Rebuttal Fig. 6). Although the exact proportion of significantly down-regulated genes varies, likely reflecting differences in replicate concordance, the MA-plots reveal that IWS1 and RTF1 depletion yield similar degrees of down-regulation, whereas SPT6 and FACT depletion induce a more pronounced downregulation, but also lead to more dispersion in RNA synthesis. Taken together, these results further support the role of IWS1 as a key elongation factor in Pol II transcription regulation.

Rebuttal Fig. 5. Boxplots show fold changes in RNA synthesis in expressed genes after 1 h IWS1 ($n=10147$), 1 h RTF1 ($n=11785$), 4 h SPT6 ($n=14002$) and 4 h FACT ($n=8969$) depletions. The thickened line represents the median and the hinges represent the first and third quartiles. The notches stretch to 1.58-times the interquartile range, divided by the square root of the sample size. Whiskers extend to 1.5-times the interquartile range from the hinge, outliers are not shown. p -values were determined using one-sample two-sided Wilcoxon test ($\mu = 0$), and shown as: ns = $P > 0.05$, * = $P < 0.05$, ** = $P < 0.01$, *** = $P < 0.001$, **** = $P < 0.0001$.

Rebuttal Fig. 6. MA plots showing differential gene expression analysis of TT-seq data after 1 h IWS1 ($n=10147$), 1 h RTF1 ($n=11785$), 4 h SPT6 ($n=14002$) and 4 h FACT ($n=8969$) depletions with $FDR < 0.05$. Significantly differentially expressed genes are shown in red (up-regulated) and green (down-regulated).

14. Figure 2 – There is a difference between the mNET-seq and the ChIP-seq results that is not commented upon: an obvious 5' accumulation of Pol II observed by mNET-seq that is not observed by ChIP-seq. The 5' effect is also evident in 2d. Please mention and discuss.

We thank the reviewer for calling attention to the differences between our mNET-seq and Pol II ChIP-seq profiles at the 5' region. As the reviewer noted, the full-gene heatmaps (TSS to 50 kb; Fig. 2b-c) show a general dIWS1-dependent Pol II buildup along gene bodies in both assays, but mask the fine structure immediately downstream of the TSS. Metagene profiles of both mNET-seq (Rebuttal Fig. 7) and Pol II ChIP-seq (Rebuttal Fig. 8) upon 1 h depletion of IWS1 show that the accumulation of Pol II happens across the gene body. As well as, boxplots of log₂ fold changes of mNET-seq counts (Fig. 2a, Extended Data Fig. 2b) and Pol II ChIP-seq counts (Extended Data Fig. 2e) show the accumulation of Pol II in the gene bodies.

We attribute the differences between Pol II ChIP-seq and mNET-seq, especially at the 5' region, to the technical differences in these assays. ChIP-seq has a resolution of 200-500 bp and also captures the pre-initiation complex. In contrast, mNET-seq captures only elongating Pol II and has single-nucleotide resolution, but sparse gene-body signal distally from the TSS. Additional factor might be different Pol II antibodies used for these assays (see Methods).

In addition, quantification of Pol II occupancy changes in the first 1 kb, middle 1 kb, and last 1 kb of each gene, reveal an accumulation of Pol II in all three gene regions, with a larger increase in the promoter-distal regions for both experimental methods (Rebuttal Fig. 9-10).

Rebuttal Fig. 7. Metagene analysis of mNET-seq coverage in non-overlapping genes (n=23036). Representations as in Rebuttal Fig. 1.

Rebuttal Fig. 8. Metagene analysis of Pol II ChIP-seq coverage in non-overlapping genes (n=23036). Representations as in Rebuttal Fig. 1

Rebuttal Fig. 9. Boxplots show fold changes in Pol II occupancy (mNET-seq) in expressed gene segments (first 1 kb, mid 1 kb, last 1 kb) after 1 h IWS1 (n=10147) depletion. Representations as in Rebuttal Fig. 5.

Rebuttal Fig. 10. Boxplots show fold changes in Pol II occupancy (Pol II ChIP-seq) in expressed gene segments (first 1 kb, mid 1 kb, last 1 kb) after 1 h IWS1 (n=10147) depletion. Representations as in Rebuttal Fig. 5.

15. Figure 3c,d – What is the barrier index and how is it calculated? It would be clearer to call this the short transcript and measure its level similar to what is done in 3d. For 3d, please explain in methods how normalized intensity was measured. Normalized to what?

We apologize for the missing description, the calculation of the barrier index and the normalization were performed as previously described (Zumer et al. 2024, PMID: 38810649). We have now included extended transcription assays performed in the presence of ELOF1 and IWS1 truncations. In these assays, the entry barrier index and full-length transcript are inversely related. We chose to omit the redundant barrier index in the revised manuscript for clarity.

16. line 142 – What happens to SETD2 recruitment after 4 hours of IWS1 depletion?

We performed chromatin proteomics following a 4-hour depletion of IWS1 and observed a modest, non-significant increase in SETD2 chromatin recruitment. Notably, changes in SETD2 levels after both 1-hour and 4-hour IWS1 depletions mirrored changes in SPT6, suggesting that SETD2 recruitment may depend more on SPT6 than on IWS1. This is further supported by our recent study that shows SPT6 interacts with SETD2 directly and helps position it for co-transcriptional methylation (Walshe et al, 2024, bioarxiv).

17. Figure 3g – SETD2 and H3K36me3 are usually recruited across transcribed sequences. Please explain how “peaks” are defined.

We show the data for the genes in panels 3e and 3f, but we wanted to be sure that we are not overlooking something by only looking at genes. Therefore, we called peaks using the broad peaks option in MACS3, 75 % of these peaks overlap genes (expressed non-overlapping), the rest include overlapping genes and likely also highly expressed enhancers. The observed changes in ChIP-seq signal in peaks (3g) match changes in genes (3e and 3f).

18. lines 151-153, Figure 3h – There are a few points regarding this result and the way it is described in the text. First, the experiment in reference 47 consisted of analysis of a single promoter under a certain set of growth conditions, a very different experiment than the proteomic analysis presented here, particularly since they were looking at a promoter region, a location where Spt6 and Iws1 are not normally found. Second, reference 29 found that there is modest decrease in Spt6 recruitment after IWS1 (Spn1) depletion. Finally, the writing will be clearer if the sentence ending on line 153 added “after IWS1 depletion” to the end of the sentence.

We compared our findings to reference 47 because it highlights a key role of Spn1 in recruiting Spt6 to specific promoter regions in yeast, providing a useful contrast to the global chromatin proteomic analysis we performed, illustrating differences between yeast and human systems. We have removed the original sentence and section relating to reference 47 and now include SPT6 with the other elongation factors in Fig. 5d and Extended Data Fig. 11a showing changes in chromatin proteomics data upon 1 h and 4 h depletions of IWS1, respectively.

19. In methods, it is stated that cells were treated for one hour for TT-seq, but this information is not provided for mNET-seq or ChIP-seq. Also, the number of replicates is not stated for ChIP-seq.

Thank you for pointing this out, all experiments in cells were performed with 1 h treatment, except the H3K36me3 ChIP-seq and chromatin proteomics, which include an additional 4 h timepoint. We added this information and the number of ChIP-seq replicates in the Methods section.

20. Please provide information on the reproducibility between the replicates for the TT-seq, mNET-seq, and ChIP-seq. Also, please clarify what is presented in the TT-seq, mNET-seq, and ChIP-seq figures. Are we looking at one of the two replicates, or merged datasets?

We included correlation plots for all the experiments to show that the data is reproducible. The data presented in the metagene profiles and heatmaps is the combined signal from two replicates, which was done as described in the methods section lines 940-950 after we checked the concordance between replicates was high by checking the correlations between replicates and individual metagene profiles of each replicate.

21. Figure 3j – Please tell us the number of replicates for this experiment and include quantitative analysis of the results.

Figure 3j (now Extended Data Fig. 11i) was performed once as a qualitative experiment. We have since developed a robust co-transcriptional methylation assay that monitors incorporation of 3H SAM (Walshe et al., 2024, bioarxiv). Results from 3x replicates are shown below and confirm the result of our qualitative western blot.

Rebuttal Fig. 11. Quantitative co-transcriptional methylation assays. Co-transcriptional methylation assays (under identical conditions to Figure 3j) were performed on chromatinized DNA templates containing four tandem Widom 601 sequences in the presence or absence of SETD2, IWS1, or transcription. Reactions were performed in triplicate. Statistical analysis between conditions performed using an unpaired t-test. Each point reflects one replicate (N=3), depicted as mean \pm s.d.

22. lines 168-169 – If the change in H3K36me3 is a result of altered transcription than this would be evident in a scatterplot of the change in TT-seq vs. the change in H3K36me3. This information should be presented.

We thank the reviewer for this suggestion, we created scatter plots comparing the changes in TT-seq (RNA synthesis) and H3K36me3 levels at 1h and 4h (Rebuttal Fig. 12). The correlation between the two measurements is positive but modest and slightly increases with longer depletion time. The fold-changes in methylation are small, which is in line with the expected slow turnover of this modification (PMID: 25774516).

Moreover, when we sort genes into quartiles based on fold-change in RNA synthesis (TT-seq) for depleted vs. control we observe that the most downregulated genes (Q1) also show a greater reduction in H3K36me3 levels and correspondingly the least downregulated genes (Q4), have a smaller reduction in H3K36me3 levels (Rebuttal Fig. 13). Taken together, these analyses support our model that transcription loss upon IWS1 depletion is causal for the decrease in H3K36me3.

Rebuttal Fig. 12. Correlation of log₂-transformed fold changes RNA synthesis (TT-seq) and H3K36me3 levels (ChIP-seq) of genes in IWS1-depleted cells compared to control cells. Spearman's *r* and *p*-values are shown for each scatterplot.

Rebuttal Fig. 13. Boxplots of log₂-transformed fold changes of RNA synthesis (TT-seq) or H3K36me3 levels (ChIP-seq) upon 1 or 4 hour depletion of IWS1 grouped by RNA-synthesis change into quartiles (Q1-Q4). P-values were determined with Wilcoxon test.

23. line 421 – change “initially” to “initial”

Thank you for pointing this out. We revised “initially” to “initial” as suggested.

Reviewer #2 (Remarks to the Author):

This study demonstrates that IWS1 plays a direct role in transcription elongation by supporting RNA polymerase II (RNAP2) activity. Its rapid depletion reduces elongation velocity and has secondary effects on H3K36me3 levels without affecting the recruitment of the H3K36me3 writer, SETD2. IWS1 is an exciting target, known to directly interact with and regulate RNAP2-mediated transcription. While the authors' approach is innovative and holds significant potential to uncover key dependencies in RNAP2 regulation, the manuscript does not meet the standards I would expect from this lab. Specifically, the genome-wide data analyses are preliminary and lack robust statistical support. Additionally, mechanistic insights into how IWS1 regulates RNAP2 activity and impacts H3K36me3 deposition or turnover are limited. Addressing this issue would provide the novelty currently lacking.

We thank the reviewer for the in-depth review, we have addressed every specific point in our revision and provide additional statistical analyses (see point-to-point response below for details). Regarding the lack of mechanistic insight, we respectfully disagree with the reviewer. Nevertheless, we have extended our study and provide additional data and molecular insight into the role of IWS1 in transcription. Specifically, we have added new results, which include a structure of the activated elongation complex with IWS1 and ELOF1, additional biochemical assays and new cell-based experiments, which led the reorganization of the entire manuscript with many changes in the manuscript text and figures. Importantly, our conclusions that IWS1 directly stimulates Pol II activity and that the decrease in H3K36me3 are a secondary effect of perturbed elongation remain unchanged, and we now provide molecular insight into the role of IWS1 in elongation.

However, deposition and turnover of H3K36me3 is out of scope for the current study, which investigates the role of IWS1 in RNA Pol II function and also shows that transcription elongation is needed to maintain H3K36me3 levels. We have reported on H3K36me3 deposition in a separate study (Walshe et al, 2024, bioarxiv).

Several major points are outlined below:

Major Comments:

1. The authors did not provide a pre-release token for their GEO data, making their genome-wide data

inaccessible to reviewers. As a result, it is not possible to assess the quality of the genomic data and evaluate their data interpretation.

We have provided the reviewer access tokens for both the NGS and MS data in the online manuscript submission form, we kindly ask the editorial team to forward these to the reviewer.

NGS datasets generated in this study have been deposited in NCBI's Gene Expression Omnibus database (GEO) and are accessible through GEO Series accession numbers GSE276547, GSE276549, GSE276550. The mass spectrometry proteomics data have been deposited to the ProteomeXchange Consortium via the PRIDE (59) partner repository with the dataset identifier PXD056248.

The NGS sequencing data have been uploaded to GEO and are accessible to the reviewers (link, accession, token)

<https://www.ncbi.nlm.nih.gov/geo/query/acc.cgi?acc=GSE276550> GSE276550 etqxeokwrfupncf

<https://www.ncbi.nlm.nih.gov/geo/query/acc.cgi?acc=GSE276549> GSE276549 ghmbokoczhqplst

<https://www.ncbi.nlm.nih.gov/geo/query/acc.cgi?acc=GSE276547> GSE276547 uvstckoglbtyxwz

The mass spectrometry proteomics data have been deposited to the ProteomeXchange Consortium via the PRIDE partner repository with the dataset identifier PXD056248.

The account for reviewer is:

Username: reviewer_pxd056248@ebi.ac.uk

Password: 2Uyl9OA34MaU

2. The statement in the abstract that "whether IWS1 has a role in RNA polymerase (Pol) II function remains unclear" overlooks previous studies that clearly demonstrate IWS1's significant role in RNAP2 regulation (PMIDs: 32941642, 34822292, 17234882, 29300974, 38559248, 33888559, and others).

We thank the reviewer for raising this point and agree the previous studies have implicated IWS1 in the regulation of PolII, however we strictly distinguish between factors that regulate the activity of RNA Pol II directly or indirectly. The first represent factors that interact with the RNA Pol II enzyme and affect its function - NTP incorporation or endonucleolytic cleavage, with examples of this being RTF1, which stimulates transcription by interacting with components of the RNA Pol II active site, TFIIIS, which stimulates cleavage, and NELF, which stabilizes the backtracked state to decrease activity. We have edited the text to clarify our point.

Taking the above defined roles in Pol II function, no study to date has reported on IWS1 controlling the activity of RNA Pol II directly. The studies referenced indeed show that IWS1/Spn1 loss or mutation leads to changes in gene expression in the case of mutant IWS1 and loss of normal expression or changes in chromatin structure in the case of Spn1, but none provide evidence of IWS1/Spn1 regulating the activity of RNA Pol II directly. We also cited and discussed the above referenced studies in our manuscript, except a preprint from July 2024 (PMID: 38559248), which we now included in the revision.

3. It is critical that the authors validate the targeted protein degradation of IWS1 using an IWS1-specific antibody. This is particularly important because the study relies on a CRISPR/Cas9 dTAG knock-in cell line, and it is crucial to confirm that no endogenous (untagged) IWS1 remains after the knock-in. Currently, the depletion of the IWS1-HA-HA-dTAG protein product is assessed using an HA antibody, which detects only the tagged IWS1 and cannot rule out the presence of endogenous IWS1.

We thank the reviewer for the suggestion and we now included validations of the tagging in the supplementary data of the manuscript (Extended Data Figure 1a). The cell line was validated to be homozygous (all copies of IWS1 are tagged) by genotyping PCR and Sanger sequencing of the locus. Additionally, we included WBs with anti-IWS1 antibody (Extended Data Figure 1c).

4. This paper presents itself as the first to reveal the role of IWS1 in transcription but does not extend beyond what has already been shown. For example, prior studies demonstrated an acute role for the yeast IWS1 homolog Spn1 in normal transcription, splicing of ribosomal protein transcripts, and proper localization of H3K36me3 along coding genes (PMID: 32941642). The rapid degradation of SPT6, an interaction partner of IWS1 that recruits it to RNAP2, highlighted its crucial role in

transcription termination (PMID: 34233157). Additionally, another study showed that mutations in IWS1 disrupting its interaction with H3K36me3 readers result in defects in releasing paused RNAP2 into productive elongation (PMID: 34822292). Does IWS1 degradation similarly affect transcription start site selection (TSS), transcription termination, or the release of promoter-proximal paused RNAP2 into elongation? The authors have already generated datasets that could address these questions.

We cite and discuss all the referenced papers in the introduction with the exception of the SPT6 study (PMID: 34233157), which does not contain data on IWS1. In contrast to the referenced IWS1/Spn1 papers we investigated nascent transcription – RNA synthesis that cannot be assessed with RNA-seq as performed in the referenced studies. RNA-seq measures total levels of RNA in a cell, which are dependent on both synthesis and degradation. Our main finding is that IWS1 affects RNA synthesis directly and not through interaction partners like SETD2 and other TND- or TIM-containing factors as suggested previously. Finally, we also show that the C-terminal region of IWS1 can stimulate Pol II transcription in vitro.

To examine whether levels of TND- or TIM-containing factors are also changed upon IWS1 depletion we checked the levels of these in our chromatin proteomics data (Rebuttal Fig. 14). We observed that out of known TND- or TIM-containing factors (PMID: 34822292), only the chromatin association of HRP2 is significantly altered upon depletion of IWS1. In contrast to IWS1, which is an essential gene in yeast and mouse (PMIDs: 35977387, 30208029, 30846735), a homozygous knockout mouse for HRP2 is viable (PMID: 33477970). This indicates a distinct HRP2-independent role for IWS1/Spn1 in transcription, but does not exclude a role of IWS1 as an interaction hub for H3K36me3 readers as noted by the reviewer.

Rebuttal Fig. 14. Fold changes in levels of Pol II and its associated factors in the chromatin fraction after 1 h depletion of IWS1. Each dot represents a factor/subunit, where colors and labels denote significant changes, non-significant changes are shown in grey. Horizontal line represents average log2 fold-change of each subgroup. P-values were assessed using Student's t-test.

Finally, we observe a global reduction in RNA synthesis upon IWS1 depletion with 74% of expressed genes being significantly downregulated (7504 out of 10147, $\text{padj} < 0.05$), whereas the IWS1 mutant for H3K36me3-reader-binding only affects a small subset (~400 genes downregulated and ~600 upregulated; PMID: 34822292) pointing towards a likely gene-specific scope for this indirect role of IWS1 in transcription.

5. The authors observe that while global SETD2 occupancy remains unchanged, H3K36me3 levels are reduced after IWS1 degradation. The authors conclude that the primary effect of IWS1 depletion is on transcription elongation, which subsequently has a secondary effect on H3K36me3 deposition. This observation is intriguing; however, it remains unclear how the observed reduction in H3K36me3 levels occurs within such a short timeframe. Turnover of H3K36me3 after SETD2 inhibition is a lengthy process, typically taking hours to days. How does the transcriptional disruption caused by IWS1 depletion lead to accelerated turnover of this histone mark? What mechanism do the authors propose to explain this rapid reduction in H3K36me3?

We thank the reviewer for this interesting point. We looked into this more systematically by performing ChIP-seq for H3K36me3 upon rapid depletion of elongation factors that affect transcription either directly, such as RTF1 or indirectly, such as SPT6. While we did not directly probe demethylase activity, we observe that rapid depletion of RTF1 or SPT6 both cause a comparable reduction in H3K36me3 (Fig. 5g-j), confirming that maintenance of this mark is dependent on ongoing transcription elongation and that turnover is indeed observed upon 1 hour of transcription perturbation. However, dissection of the precise mechanism of histone turnover lies beyond the scope of the present study.

6. The authors claim that H3K36me3 is a secondary effect of transcriptional deregulation. Is there a relationship between the observed transcriptional downregulation and the decrease in H3K36me3 levels at individual gene bodies? The authors need to show this to support their claim.

We thank the reviewer for the suggestion, to check this, we divided genes into quartiles based on their transcriptional log₂ fold-changes (TT-seq), where “top” and “bottom changed” correspond to the top and bottom 25% of the significantly down-regulated genes with largest and smallest -log₂ fold-change, respectively. The top changed genes also have a more pronounced reduction in H3K36me3 after 1 hour of IWS1 depletion (Rebuttal Fig. 15) and the trend is the same after longer depletion (4 hours), but the difference and decrease are more pronounced.

Rebuttal Fig. 15. Boxplots of log₂-transformed fold changes of RNA synthesis (TT-seq) or H3K35me3 levels (ChIP-seq) upon 1 or 4 hour depletion of IWS1 for top and bottom 25% genes based on change in RNA synthesis. *p*-values were determined with Wilcoxon test.

Furthermore, when we compare per gene changes in TT-seq (transcription) and H3K36me3, we observe a modest positive correlation between the two measurements (Rebuttal Fig. 16). The changes in methylation are small, which is in line with the expected slow turnover of H3K36me3. Taken together these analyses support our model.

Rebuttal Fig. 16. Correlation of log₂-transformed fold changes RNA synthesis (TT-seq) and H3K36me₃ levels (ChIP-seq) of genes in IWS1-depleted cells compared to control cells. Spearman's *r* and *p*-values are shown for each scatterplot.

7. The authors should provide a more comprehensive analysis of their ChIP-seq data beyond the box plots currently presented. Specifically, they should provide separate heatmaps for each replicate to clearly show the loss of H3K36me₃ and the retention of SETD2 across the relevant genomic regions. Additionally, example browser tracks should be included to visually demonstrate these changes. Without these additional analyses, it will be difficult for readers to fully trust the conclusions.

We generated heatmaps for H3K36me₃ ChIP-seq for genic regions and included them to the manuscript. Additionally, we performed chromatin proteomics after 4-hour depletion of IWS1 and did not observe a significant change in chromatin recruitment of SETD2 (Extended Data Fig. 12a) similar to 1 hour depletion (Fig. 5a). Here, we show browser tracks of H3K36me₃ ChIP-seq after RTF1, SPT6, and IWS1 depletions, which clearly show the decrease of this histone mark upon depletion of elongation factors for two genes (Rebuttal Fig. 17). The decrease is very pronounced after depletion of SPT6, but this is likely due its role in position SETD2 for co-transcriptional methylation (Walshe et al, 2024; bioarxiv, Markert et al., 2025; PMID: 39666822, Kujirai et al., 2024; bioarxiv). We removed the SETD2 ChIP-seq data because we were unable to reproduce our previous enrichment levels with a new lot of SETD2 antibody. Instead, we included chromatin proteomics for a second longer timepoint (4 hours) similarly showing no significant change in SETD2 recruitment.

Rebuttal Fig. 17. iGV browser tracks of H3K36me₃ ChIP-seq coverages upon depletion of RTF1 (1h), SPT6 (4h), IWS1 (both 1 h and 4 h depletions), and SETD2 ChIP-seq upon 1 h IWS1 depletion. Coverages of control (gray) and dTAG7 (colored) treated samples are overlaid. Here two example genes (MYC and GAPDH) are provided.

8. The authors do not provide statistical measures to support many of their claims. Of particular concern is the main claim that rapid depletion of IWS1 leads to an increase in RNAP2 occupancy at genes and a decrease in elongation velocity. This central finding is not appropriately supported by statistical analysis. This issue should be addressed throughout the manuscript to ensure that all the authors' claims are substantiated.

In our initial submission, we included statistical analysis to obtain *p*-values for all direct comparisons of treated and control in all boxplot representations of data and we plot all metagene profiles with confidence intervals for the bootstrapped mean. To alleviate the reviewers' concerns, we have now added *p*-values to boxplots showing log-fold changes (one-sample two-sided Wilcoxon test, $\mu=0$).

In general, we prefer heatmaps for visualization of changes in Pol II occupancy and elongation velocity, because they provide spatial per gene representation of signal. For thoroughness, we now provide an additional quantitative comparison to provide *p*-values for the changes in Pol II elongation velocity (Fig. 2e). The quantitative comparison of Pol II occupancy (mNET-seq) was already provided

in Fig. 2a and Extended Data Fig. 2b. We added quantitative comparison of Pol II occupancy based on Pol II ChIP-seq data in Extended Data Fig. 2e.

9. Figure 2A must include control data to compare the RNAP2 occupancy before IWS1 degradation. As it currently stands, the figure is incomplete.

Fig 2A shows fold-change of treated/control, which includes information about the control. We now added p-values to all boxplots showing log-fold changes (one-sample two-sided Wilcoxon test, $\mu=0$).

Reviewer #3 (Remarks to the Author):

This is a well done manuscript which shows that the mammalian IWS1 protein stimulates transcription in vivo and in vitro. The experiments are well done and the conclusions are solid. It is a short report and conclusive.

We thank the reviewer for the evaluation of the quality of our experimental work.

The difficulty I have with the manuscript is that it does not provide significant novel information beyond what is known previously about IWS1 and its yeast homolog. It lacks mechanistic insight other than to suggest its activity is not mediated through recruitment or activity of SETD2. While these confirming results will be of interest to the elongation field it does not provide enough novel insights for publication in Nature Communications. The current work could serve as an important part of a longer paper providing additional data addressing mechanisms or it could be published in a more specialized journal.

We have added new results, which include a structure of the activated elongation complex with IWS1 and ELOF1, additional biochemical assays and new cell-based experiments, which led the reorganization of the entire manuscript with many changes in the manuscript text and figures. Importantly, our conclusions that IWS1 directly stimulates Pol II activity and that the decrease in H3K36me3 are a secondary effect of perturbed elongation remain unchanged, and we now provide molecular insight into the role of IWS1 in elongation.

Specifically, we have now extended our study and we now provide:

- A high resolution cryo-EM structure of the human activated elongation complex (Fig. 4), which includes IWS1 and ELOF1 that defines the C-terminal region as critical for transcription stimulation.
- Biochemical assays demonstrating that the disordered C-terminal region of IWS1, synergizing with ELOF1, directly promotes Pol II passage through a nucleosome barrier (Fig. 5a–c). This provides functional evidence for the structural observations and establishes a direct stimulatory role for IWS1 in elongation.
- Comparative rapid-depletion experiments for RTF1 and SPT6 (Fig. 5g–j), showing that loss of these elongation factors yields a similar reduction in H3K36me3 as observed after IWS1 depletion. This demonstrates that H3K36 trimethylation is supported by elongation, and not specifically IWS1.

Together, these new data deliver a coherent, molecular-level mechanism for role of IWS1 in transcription elongation—far beyond merely ruling out SETD2 recruitment—and firmly justify publication.

REVIEWER COMMENTS

Reviewer #1 (Remarks to the Author):

The authors have done a nice job of responding to the reviewers' comments. The addition of the structural studies adds interesting new results. I think that the manuscript now merits publication.

We thank the reviewer for their recommendation.

Reviewer #2 (Remarks to the Author):

The authors have addressed some, but not all, concerns from the initial review. Several major issues remain, primarily because key analyses were conducted for the reviewers but not incorporated into the manuscript. The addition of the structural model raises further concerns, particularly due to its reliance on unvalidated AlphaFold predictions and the weak integration of the new data into the overall narrative. Please see detailed comments below:

1) The authors performed a substantial number of analyses to address the reviewers' concerns, including those raised by me and Reviewer 1. However, many of these analyses were shared only as rebuttal figures in the response to reviewers and were not incorporated into the manuscript. This approach does not adequately resolve the issues raised. While not every analysis needs to be included, many are directly relevant and should be presented in the manuscript, as readers are likely to raise similar questions.

We thank the reviewer for their valuable feedback regarding the incorporation of additional analyses into the manuscript. We have now carefully reviewed each suggested addition. Specifically, to address the points raised by both Reviewer 1 and Reviewer 2, we have now incorporated the plot showing the comparison of H3K36me3 levels at genes most and least affected by IWS1 depletion based on TT-seq data into the manuscript as Extended Data Fig. 11j. This figure clearly demonstrates that genes exhibiting the most significant transcriptional decrease also show a corresponding reduction in H3K36me3. We also included the analysis requested in point 5, see below for details.

While we have integrated key analyses that directly address novel insights or resolve specific concerns, we decided not to incorporate all presented analyses, as some provided redundant information or did not add substantially new insights beyond what is already presented in the main figures and text. Please note that our detailed responses to all reviewer comments will be published alongside the manuscript by the journal, so all analyses presented here will be accessible to the readers.

2) The authors present a new structural analysis extending existing models by identifying additional interfaces of IWS1, including an “elongation complex scaffold domain” proposed to stimulate RNA Polymerase II. Since IWS1's disordered regions are established in the literature as interaction scaffold for multiple elongation factors, the novelty of this interface relies on the identification and structural analysis of the interfaces. However, this disordered domain is modeled by fitting an AlphaFold predictions into unassigned cryo-EM

density without experimental validation—neither mutagenesis nor structural mapping (e.g., X-ray crystallography, NMR, XL-MS) has been performed. While the authors demonstrate that deleting the entire domain impairs Pol II stimulation, the individual contacts remain unvalidated. Experimental confirmation of these predicted interfaces is essential to substantiate the structural claims. Without such validation, the proposed scaffold domain remains speculative and should not be presented as a confirmed structural feature.

We thank the reviewer for their helpful suggestions and apologize for any confusion. To clarify:

In Figure 4d,e, we show cryo-EM densities from MAPS 8 and 9, resolved at 2.65 Å and 2.92 Å, respectively. These densities are of sufficient quality to unambiguously assign amino acid composition and register. Thus, contrary to the reviewer's suggestion, the densities are not "unassigned." AlphaFold 3 was used solely to assist model building—not to identify the protein factors. These assignments were further validated by in vitro biochemical data confirming the functional relevance of the observed interactions. We have added a clarifying statement to the Methods section regarding the role of AlphaFold in model building.

The TND Interaction Motifs (TIMs) within the N-terminal disordered region of IWS1 have been previously reported to bind elongation factors (Cermakova et al., Science 2021). However, our RNA extension assays demonstrate that this region does not directly stimulate RNA polymerase II.

As described in the Results, we collectively refer to the four IWS1 regions that interact with Pol II, DNA, and elongation factors as the elongation complex scaffold (ECS). We use the term scaffold to emphasize that these regions coordinate interactions among multiple components, as clearly supported by our high-resolution cryo-EM maps.

Importantly, our RNA extension assays demonstrate only the C-terminal residues of the ECS domain (Interaction Site 4) stimulate Pol II activity. This narrowed the functionally relevant region to the 40 C-terminal amino acids. We have revised the Results section and adjusted Figure 5, to more precisely describe the functional boundaries of this region.

3) The authors have added new structural data to enhance the manuscript; however, ELOF1 is not well integrated into the overall narrative. The study is primarily centered on IWS1, but the inclusion of ELOF1 disrupts the focus and weakens the framing. Does ELOF1 have a unique functional relationship with IWS1 that distinguishes it from other elongation factors? If so, this should be clearly articulated. Why was ELOF1 chosen as a focal point over other well-characterized elongation factors that also interact with IWS1 such as TFIIS, SPT6, or SPT5?

As noted in both the Introduction and Results, the yeast homologues E1f1 and Spn1 exhibit overlapping gene occupancy profiles (Mayer et al., Nat Struct Mol Biol 2010) and bind in close spatial proximity on RNA polymerase II (Ehara et al., Science 2019). This provided a

strong rationale to investigate the stimulatory role of IWS1 in the presence and absence of ELOF1. Our high-resolution cryo-EM analysis of this complex revealed a previously uncharacterized interaction between IWS1 and ELOF1. The inclusion of these data directly addresses the initial concern raised by Reviewer 3 regarding the scope of the study.

Our RNA extension assays clearly demonstrate a synergistic effect of IWS1 and ELOF1 in stimulating Pol II activity. While the question of whether this synergy is unique is indeed compelling, it falls outside the scope of the present study, which—as the reviewer correctly notes—is focused primarily on the role of IWS1.

4) The authors refer to both the IWS1 “core domain” and the “TND domain” throughout the manuscript. However, IWS1 has only one folded domain—commonly known as the TND—which includes an additional HEAT repeat. While a recent yeast transcription complex study used the term “core domain,” the majority of the literature refers to this region as the TND, sometimes specifying the HEAT repeat. Switching between two different terms without clear explanation is misleading and may imply the presence of two separate domains. The authors should clarify this terminology and consider adopting the more widely accepted nomenclature, particularly since the manuscript includes structural data and the terms “TND” and “HEAT repeat” accurately reflect the domain’s architecture.

We thank the reviewer for clarifying this point. We note that the term “TND” was used only in reference to the TND-interaction motifs (TIMs), to distinguish these motifs from the structured helical bundle that contains the TND. As noted, the structured helical bundle of IWS1 consists of two components. To better reflect this organization, we opted to use the term “core.” We have now added a clarifying statement in the Introduction to avoid confusion.

5) The explanation of Figure 2A does not address my concern. Presenting a bar graph with a single column showing fold change, p-value, and standard deviation—without including the reference or baseline group—is problematic. Since fold change is inherently comparative, omitting the baseline obscures essential context and limits the interpretability of the results. This is particularly important in genomics, where subtle differences can yield statistically significant p-values without reflecting biologically meaningful effects.

We thank the reviewer for their feedback. In response to it, we have added Extended Data Fig. 2B, which displays box plots of the normalized mNET-seq counts for both the control and dIWS1 samples separately. This figure complements Fig. 2A, which shows the fold change, by allowing for a direct visualization of the distribution of the underlying data for each condition. As demonstrated in Extended Data Fig. 2B, the difference in normalized counts between the control and dIWS1 samples is statistically significant, further supporting our findings presented in Fig. 2A. We believe this addition enhances the clarity and interpretability of our findings.

6) Minor issue: The authors clearly state that IWS1 is N-terminally tagged with dTAG; however, the labeling throughout the manuscript—especially in the figures—is potentially confusing. For instance, Figure 1a lacks sufficient resolution to show the orientation of the IWS1 gene, which may give the impression that the tag is C-terminal. Additionally, the

notation "IWS1-dTAG(HA)" suggests a C-terminal tag. I recommend clearer and more consistent labeling to prevent misinterpretation.

We thank the reviewer for their feedback. We have revised the manuscript and figures to ensure consistency and clarity regarding the construct used. All instances of "IWS1-dTAG" have now been updated to "dTAG-IWS1" throughout the text and figures. Furthermore, we have updated Fig. 1A to make the schematic clearer. It now explicitly illustrates that the dTAG tag was added to the N-terminus of IWS1 gene.

Reviewer #3 (Remarks to the Author):

my concern with the original manuscript was the limit of novel results that was reported. The new structural and biochemical experiments included in the revised manuscript extend the story and provide additional novel mechanistic insights. The revised manuscript is suitable for publication.

We thank the reviewer for their recommendation.